# Carm1-arginine methylation of the transcription factor C/EBPα regulates transdifferentiation velocity

Guillem Torcal Garcia[1,2†], Elisabeth Kowenz-Leutz[3†], Tian V Tian[1,2,4†], Antonis Klonizakis[1,2†], Jonathan Lerner[5], Luisa De Andres-Aguayo[1,2], Valeriia Sapozhnikova[3], Clara Berenguer[1,2], Marcos Plana Carmona[1,2], Maria Vila Casadesus[1,2,4], Romain Bulteau[6], Mirko Francesconi[6], Sandra Peiro[4], Philipp Mertins[3], Kenneth Zaret[5], Achim Leutz[3‡], Thomas Graf[1,2*‡]

[1]Centre for Genomic Regulation (CRG), The Barcelona Institute of Science and Technology (BIST), Barcelona, Spain; [2]Universitat Pompeu Fabra (UPF), Barcelona, Spain; [3]Max Delbrück Center for Molecular Medicine in the Helmholtz Association, Berlin, Germany; [4]Vall d'Hebron Institute of Oncology (VHIO), Barcelona, Spain; [5]Perelman School of Medicine, University of Pennsylvania, Philadelphia, United States; [6]Laboratorie de Biologie et Modélisation de la Cellule, Université de Lyon, Lyon, France

*For correspondence:
Thomas.Graf@crg.eu

†These authors contributed equally to this work
‡These authors also contributed equally to this work

**Abstract** Here, we describe how the speed of C/EBPα-induced B cell to macrophage transdifferentiation (BMT) can be regulated, using both mouse and human models. The identification of a mutant of C/EBPα (C/EBPα[R35A]) that greatly accelerates BMT helped to illuminate the mechanism. Thus, incoming C/EBPα binds to PU.1, an obligate partner expressed in B cells, leading to the release of PU.1 from B cell enhancers, chromatin closing and silencing of the B cell program. Released PU.1 redistributes to macrophage enhancers newly occupied by C/EBPα, causing chromatin opening and activation of macrophage genes. All these steps are accelerated by C/EBPα[R35A], initiated by its increased affinity for PU.1. Wild-type C/EBPα is methylated by Carm1 at arginine 35 and the enzyme's perturbations modulate BMT velocity as predicted from the observations with the mutant. Increasing the proportion of unmethylated C/EBPα in granulocyte/macrophage progenitors by inhibiting Carm1 biases the cell's differentiation toward macrophages, suggesting that cell fate decision velocity and lineage directionality are closely linked processes.

## Editor's evaluation

This important manuscript describes how the methylation of a single arginine residue in a transcription factor, C/EBPα, can alter the dynamics of cell fate transition. The study provides one of the most striking examples of transcription factor regulation by methylation and is well-executed, with compelling evidence to support the authors' claims. These findings substantially advance our understanding of a major research question.

## Introduction

The mechanisms underlying cell fate decisions during differentiation and development have long been a subject of fascination among biologists. It is widely accepted that lineage-restricted transcription factors (TFs) play a crucial role in driving cell differentiation by repressing the old gene expression program while activating new ones at bifurcation points. The balance between antagonistic TFs is

a key determinant of cell fate, with the more highly expressed factor silencing the opposing factor (*Graf and Enver, 2009*; *Orkin and Zon, 2008*). However, the precise mechanisms, such as how gene silencing and activation are coordinated, and whether factors other than TF levels influence cell fate decisions, are still largely unknown.

A powerful method for studying mechanisms of cell fate decisions is to induce lineage conversions through TF overexpression (*Graf, 2002*; *Zhou and Melton, 2008*; *Graf and Enver, 2009*). An especially efficient system for this purpose is the C/EBPα-induced transdifferentiation of B cells into macrophages, designated here as BMT (*Xie et al., 2004* ; *Figure 1A*). C/EBPα is composed of a bZip DNA-binding domain on its C-terminus and a tripartite transactivation domain on its N-terminus (*Figure 1B*; *Ramberger et al., 2021b*). During hematopoiesis, it is most highly expressed in granulocyte-macrophage progenitors (GMPs) (*Ohlsson et al., 2016*) and its absence inhibits the formation of GMPs, and monocytes/macrophages (*Heath et al., 2004*; *Ma et al., 2014*; *Zhang et al., 2004*). The C/EBPα-induced BMT requires partnership with the TF PU.1, which participates in regulatory networks of both B/T cells and in myeloid cells (*Rothenberg, 2014*; *Laiosa et al., 2006*; *Heinz et al., 2010*; *Arinobu et al., 2007*; *Leddin et al., 2011*; *Singh et al., 1999*).

Post-translational modifications of TFs can have a significant impact on their structure, subcellular localization, interactions with other proteins and activity (*Deribe et al., 2010*; *Torcal Garcia and Graf, 2021*). Recently, specific protein arginine methyltransferases (PRMTs) that effect arginine modifications resulting in asymmetric or symmetric di- or mono-methylation have gained prominence (*Wu et al., 2021*; *Guo et al., 2010*; *Kawabe et al., 2012*; *Kowenz-Leutz et al., 2010*). In the context of cell fate decisions, Carm1 (Prmt4) is of particular interest as it is crucial for early embryo development, muscle regeneration, adipogenesis, and cancer (*Kawabe et al., 2012*; *Kim et al., 2010*; *Li et al., 2013*; *Torres-Padilla et al., 2007*; *Yadav et al., 2008*).

## Results

Our study revealed that Carm1-mediated methylation of a specific arginine located in the intrinsically disordered transcription activation domain of C/EBPα plays a critical role in regulating the speed of transdifferentiation induced by this factor. In its mutated or unmethylated form C/EBPα induces BMT at a dramatically increased velocity, initiated by a higher interaction affinity for its partner TF PU.1. This triggers the acceleration of chromatin accessibility changes and the redistribution of PU.1 from B- to M-GREs. The methyltransferase involved in the methylation of C/EBPa is Carm1, and it plays a role in determining the lineage directionality of GMPs.

### A mutation of C/EBPα at arginine 35 accelerates BMT

In ongoing efforts to identify mutants of C/EBPα in post-translationally modified amino acids that affect BMT (*Figure 1A*), we used retroviral constructs with C/EBPα fused to ER and inducible by 17ß-estradiol (E2). Rather than finding loss-of-function mutations as expected, we identified one that dramatically accelerates the pace of BMT (*Figure 1B and C*). This mutant (C/EBPα) contains alanine replacements of three evolutionarily conserved arginines located in the N-terminal TEI and TEII transcription activation elements of C/EBPα.

The individual contributions of the three arginines to the cells' phenotype were studied by testing the effect of single alanine replacements for R12, R35, and R86, again using E2-inducible constructs. Results showed that C/EBPα$^{R35A}$ recapitulated the phenotype induced by C/EBPα, while R12A and R86A essentially displayed no effect (*Figure 1D* and *Figure 1—figure supplement 1A*). R35A cells induced for 120 hr resembled normal macrophages, showing extensive F-actin filaments and eccentric nuclei, as well as high phagocytic capacity (*Figure 1—figure supplement 1B, C*). These findings indicate that replacing arginine 35 with alanine in C/EBPα dramatically increases the speed of induced BMT, generating macrophage-like cells similar to those obtained with the wild-type.

### C/EBPα$^{R35A}$ hastens early gene expression changes of lineage-associated genes

To investigate the impact of C/EBPα$^{R35A}$ on gene expression, we conducted RNA-sequencing (RNA-seq) on infected B cells spanning very early to late timepoints (0, 1, 3, 6, 18, and 120 hr; *Figure 1E*). The analysis revealed 11,780 genes whose expression changed by more than twofold throughout

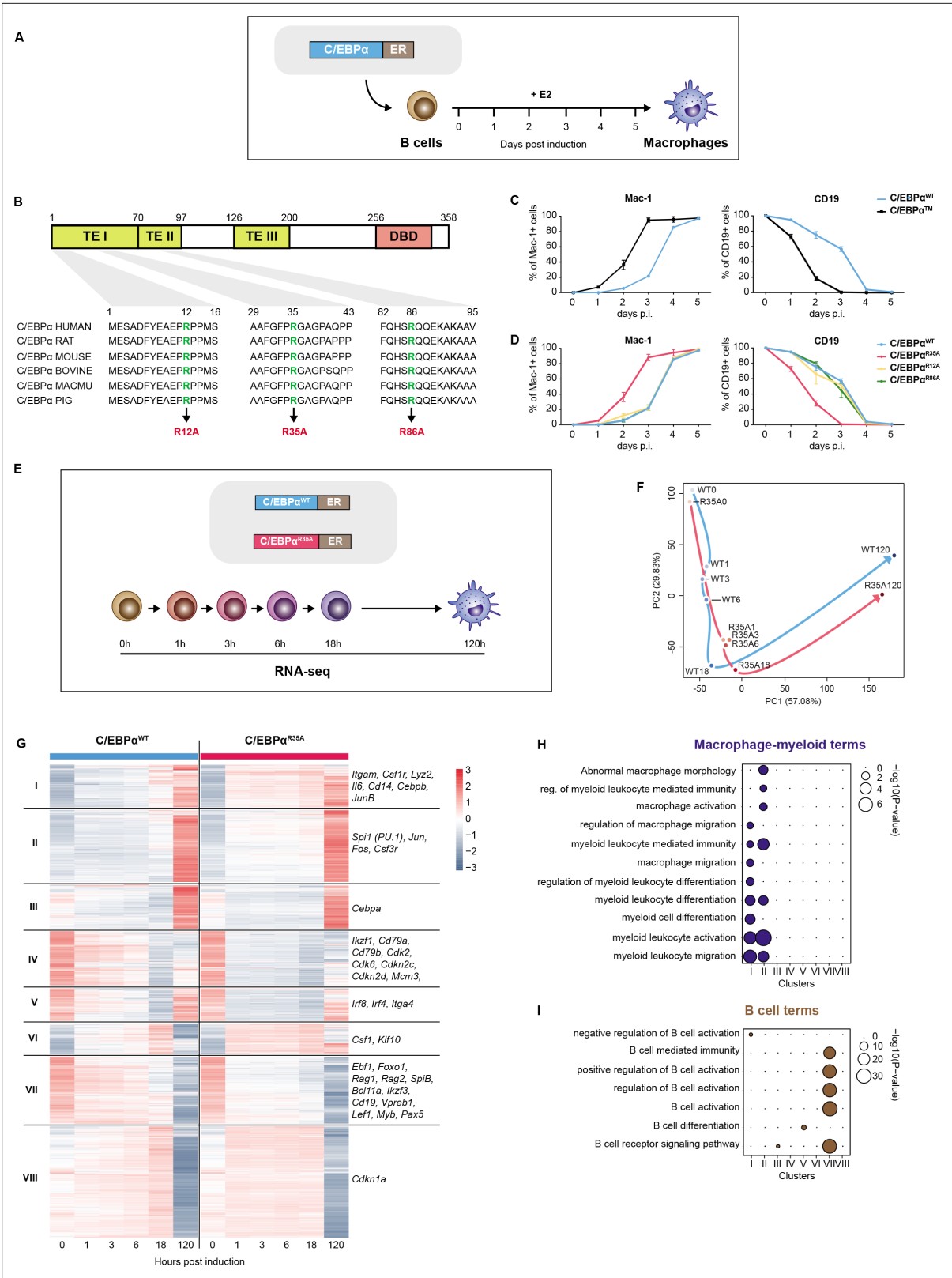

**Figure 1.** A mutation of arginine 35 in C/EBPα accelerates C/EBPα-induced B cell to macrophage transdifferentiation (BMT). (**A**) Schematics of the BMT method using primary cells. Bone marrow-derived pre-B cells infected with C/EBPα-ER-GFP retrovirus were treated with E2 to induce the factor's translocation into the nucleus, inducing a BMT within 4–5 days as assessed by Mac-1 and CD19 expression. (**B**) C/EBPα structure (TE = transactivation element; DBD = DNA-binding domain) and location of conserved arginines R12, R35, and R86. (**C**) Kinetics of BMT induced by C/EBPα wild-type (WT)

*Figure 1 continued on next page*

*Figure 1 continued*

and C/EBPα with alanine replacements of R12, R35, and R86. BMT (mean ± s.d., n=3). (**D**) As in (**C**) but comparing the effects of separate mutations in the three arginines, showing that C/EBPα^R35A is the critical residue. (**E**) Schematics of experimental approach for RNA-sequencing (RNA-seq) of B cells infected with either C/EBPα^WT-ER or C/EBPα^R35A-ER retroviral constructs induced for various timepoints. (**F**) Principal component analysis (PCA) of 11,780 differentially expressed genes during BMT (n=2). Trajectories connecting individual timepoints are visualized in blue for the wild-type and in red for the mutant. (**G**) Hierarchical clustering of differentially expressed genes with examples shown next to each cluster. (**H–I**) Gene Ontology (GO) enrichment analysis of macrophage-myeloid (**H**) and B cell (**I**) terms of the clusters from (**G**). Diameter of circles is proportional to the p-value.

The online version of this article includes the following figure supplement(s) for figure 1:

**Figure supplement 1.** Mutation of arginine 35 in C/EBPα accelerates B cell gene silencing and macrophage gene activation.

BMT in either system. Principal component analysis (PCA) showed a pronounced acceleration of gene expression changes in cells infected with the mutant virus compared to the wild-type. At just 1 hpi, the gene expression patterns in C/EBPα^R35A-infected cells were similar to those of 18 hpi wild-type-infected cells, reaching similar expression levels at 120 hpi. The largest differences in gene expression between the wild-type and mutant cells were observed between 0 and 1 hpi (*Figure 1F* and *Figure 1—figure supplement 1D*).

Hierarchical clustering of the 11,780 differentially expressed genes (DEGs) resulted in eight clusters (*Figure 1G*). Most genes in clusters I, II, IV, and VIII showed faster activation by C/EBPα^R35A, while those in clusters IV, V, and VII showed faster silencing. Clusters I and II were enriched for macrophage-myeloid-related Gene Ontology (GO) terms and included the myeloid genes *Itgam*, *Lyz2*, *Csf1r*, and *Cd14* (*Figure 1H* and *Figure 1—figure supplement 1F, G*). On the other hand, cluster VII was enriched for B cell-related GO terms and included the B cell genes *Cd19*, *Pax5*, *Ebf1*, and *Rag2* (*Figure 1I* and *Figure 1—figure supplement 1F, G*). The kinetics of macrophage- and B cell-restricted genes (*Figure 1* and *Figure 1—figure supplement 1G, H*) further highlight the acceleration of BMT induced by C/EBPα^R35A.

Through examination of thousands of differentially regulated lineage-associated genes, these results extend the distinction between C/EBPα mutant- and wild-type-induced BMT observed by FACS. Here, however, the most significant differences in gene expression were found at the earliest stages (1–3 hpi).

## C/EBPα^R35A accelerates chromatin remodeling at regulatory elements of lineage-restricted genes

To examine the chromatin remodeling accompanying induced gene expression changes during BMT, we conducted an assay for transposase-accessible chromatin sequencing (ATAC-seq) (*Buenrostro et al., 2015*) of samples collected at multiple timepoints after activation of C/EBPα^WT and C/EBPα^R35A (*Figure 2A*). Our analysis identified 91,830 regions that showed significant differences in chromatin accessibility throughout wild-type- or mutant-induced BMT, representing in their vast majority TF-bound GREs (*Buenrostro et al., 2015*). These regions were grouped into (1) faster opening GREs induced by C/EBPα^R35A, with the highest accessibility at 120 hpi (43,429 GREs); (2) faster closing GREs, most accessible at 0 hpi (36,380 GREs); and (3) transiently opening GREs, most accessible at 18hpi (12,021 GREs) that showed little differences between the two conditions (*Figure 2—figure supplement 1A,B*). PCA of the differential ATAC peaks further confirmed the acceleration of chromatin accessibility changes by C/EBPα^R35A, with the coordinates of the 1–6 hpi C/EBPα^R35A samples resembling those of the 18 hpi C/EBPα^WT sample (*Figure 2B*).

Focusing on differentially remodeled promoter regions, we found 14,233 and these were grouped into eight clusters (*Figure 2C*). Promoter-associated genes in clusters I, II, and III exhibited dramatically accelerated opening dynamics for C/EBPα^R35A, while genes in clusters VII and VIII showed accelerated closing. GO analysis revealed an enrichment of macrophage terms for cluster III, which included the macrophage-restricted genes *Itgam* and *Lyz2* (*Figure 2C*). C/EBPα^R35A already initiated chromatin opening at promoters and enhancers of *Itgam* and *Lyz2* within 1 hpi, and this increased over time much more rapidly than with C/EBPα^WT cells (*Figure 2—figure supplement 1C*). Conversely, faster closing promoters in cluster VIII were enriched for B cell-related GO terms and included the B cell genes *Cd19*, *Pax5*, and *Rag2* (*Figure 2C and E*). The kinetics of chromatin closing were likewise accelerated at B cell GREs. Strikingly, however, as exemplified for *Cd19* and *Pax5*, while mutant-induced

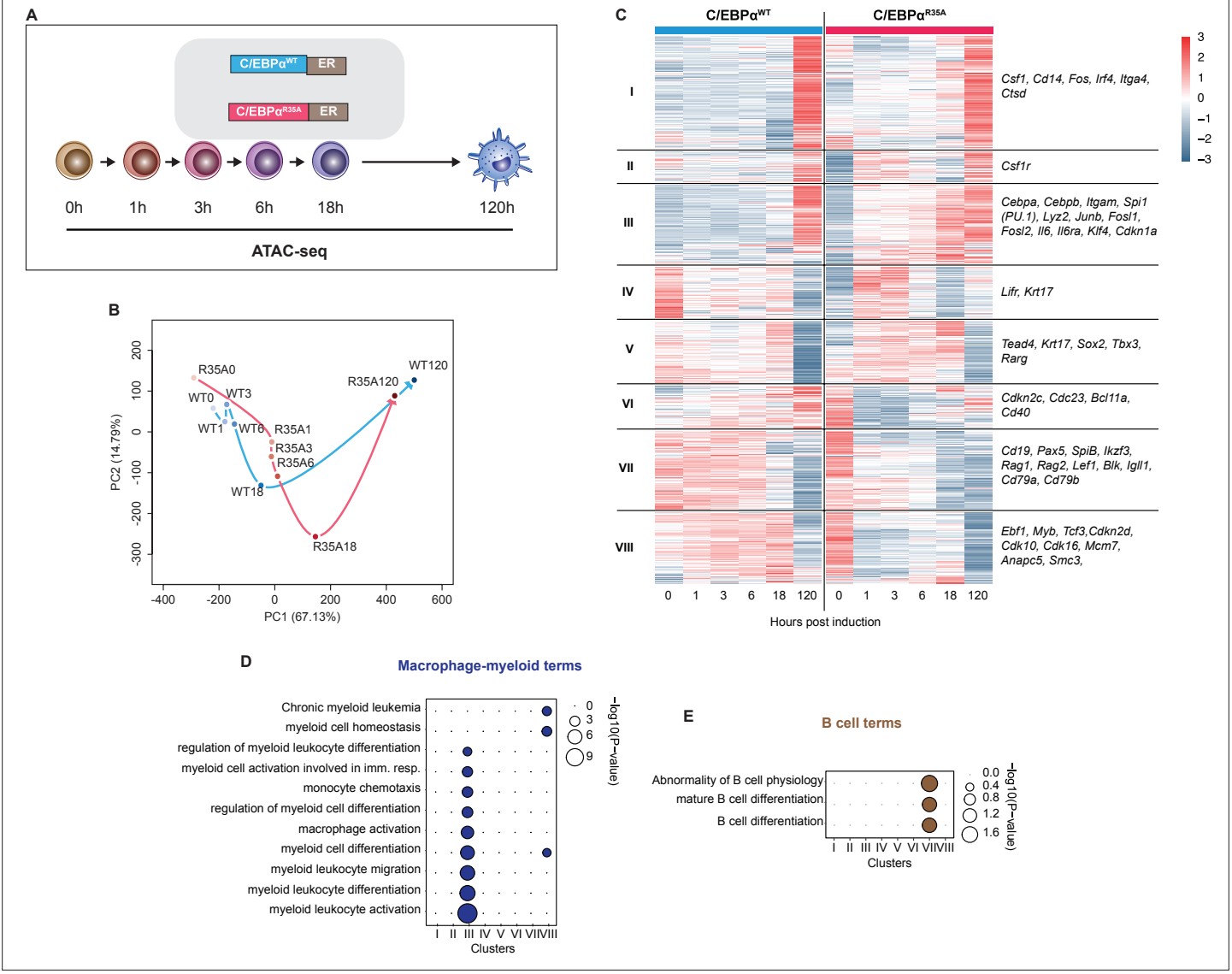

**Figure 2.** C/EBPα^R35A accelerates chromatin accessibility at gene regulatory elements of lineage-restricted genes. (**A**) Experimental approach used for chromatin accessibility profiling. B cells infected with either C/EBPα^WT-ER or C/EBPα^R35A-ER retroviral constructs (n=2 biological replicates) were induced for the indicated times and processed for assay for transposase-accessible chromatin sequencing (ATAC-seq). (**B**) Principal component analysis (PCA) of differential chromatin accessibility dynamics during B cell to macrophage transdifferentiation (BMT) induced by C/EBPα^WT or C/EBPα^R35A, based on 91,830 ATAC-seq peaks differentially called for the two conditions. Arrows connecting individual timepoints show trajectories. (**C**) Hierarchical clustering of differentially accessible promoters (14,233 peaks) with representative genes shown next to each cluster. (**D, E**) Gene Ontology analysis of macrophage-myeloid (**D**) and B cell (**E**) terms of each cluster. Diameter of circles is proportional to the p-value.

The online version of this article includes the following figure supplement(s) for figure 2:

**Figure supplement 1.** C/EBPα^R35A accelerates chromatin remodeling at regulatory elements of lineage-restricted genes.

changes occurred already at 1 hpi and progressed from there onward, the same sites showed a transient chromatin opening in wild-type-induced cells (***Figure 2—figure supplement 1D***).

Our results show that C/EBPα^R35A is more effective than C/EBPα^WT in inducing changes of chromatin closing and opening at B- and M-GREs. However, while the mutant acts immediately to close chromatin, wild-type-induced chromatin closing is preceded by a transient chromatin opening, as is also reflected by a transient increase in gene expression (***Figure 1—figure supplement 1F, G***). This is in line with the recently described Gata1-induced silencing of the *Kit* gene enhancer during erythroid cell differentiation, which becomes transiently activated (***Vermunt et al., 2023***). It is also noteworthy that despite progressive alterations in chromatin accessibility at B- and M-GREs during

BMT induction (*Figure 2—figure supplement 1D*), essentially no changes in gene expression were observed between 6–18 hpi, suggesting that chromatin remodeling is uncoupled from gene expression during this phase of the process.

## C/EBPα[35A] preferentially interacts with PU.1

To determine if the enhanced ability of the mutant to modify chromatin accessibility was due to differences in DNA binding capacity, we compared nuclear extracts of 293T transfected with C/EBPα[WT] and C/EBPα[R35A]. An electrophoretic mobility shift assay (EMSA) revealed no significant differences in overall DNA binding affinities between the two C/EBPα forms (*Figure 3—figure supplement 1A*).

We next explored whether the differences are due instead to the mutant's preferential interaction with other proteins, performing a de novo motif discovery analyses on differentially accessible GREs. Results in *Figure 3A* showed that faster and transiently opening GREs were enriched for AP-1/leucine zipper family TF motifs (c-Fos, c-Jun, and JunB), known to heterodimerize with C/EBPα and activate myeloid genes (*Cai et al., 2008*). In contrast, faster closing GREs were mostly enriched for ETS family TF motifs (Ets1, Fli1, SpiB, and Gabpa), linked to B cell lineage differentiation and function (*Eyquem et al., 2004*; *Hu et al., 2001*; *Xue et al., 2007*). Of note, the Spi1 (PU.1) motif was enriched in both the accelerated chromatin opening and closing groups (*Figure 3B* and *Figure 3—figure supplement 1B*), possibly reflecting the known dual roles of PU.1 in B cells and macrophages (*Scott et al., 1994*; *Singh et al., 1999*; *Heinz et al., 2010*; *Rothenberg, 2014*). However, no PU.1 motif enrichment was seen in the transiently opening group (*Figure 3—figure supplement 1C*).

To examine whether C/EBPα[WT] and C/EBPα[R35A] differentially interact with other proteins, we performed a proximity-dependent biotin identification assay (*Figure 3B*), which was previously used to identify high confidence C/EBPα interactors (*Ramberger et al., 2021a*). We used a mouse pre-B cell line expressing doxycycline (Dox)-inducible C/EBPα[WT] or C/EBPα[R35A] constructs fused with the biotin ligase variant TurboID (*Branon et al., 2018*). To avoid confounding interactions this cell line contained a double knockout of *Cebpa* and *Cebpb*, genes which are rapidly activated by exogenous C/EBPα (*Bussmann et al., 2009*). After Dox induction for 6 hr and biotin labeling, biotin-labeled proteins were identified by mass spectrometry. A comparison of the high confidence C/EBPα[WT] and C/EBPα[R35A] interactome revealed that C/EBPα[R35A] preferentially interacted with Spi1 (PU.1), showing a 3.6-fold increase in biotin label compared to C/EBPα[WT]. In addition, preferential but non-significant interactions of C/EBPα[R35A] were observed with Ncor1 and Ebf1 (*Figure 3B*; *Figure 3B*, source data 1), which, however, were not further studied. The observation that most shared interactors show a skewing toward the wild-type protein suggests that C/EBPα[WT] is expressed at slightly higher levels than the mutant, in line with the detection of moderately reduced levels of the mutant protein by western blot (*Figure 4—figure supplement 1C*). To further evaluate the interaction between C/EBPα[WT] and C/EBPα[R35A] with PU.1, we conducted co-immunoprecipitation (Co-IP) experiments with 293T cells co-transfected with PU.1 and either C/EBPα[WT] or C/EBPα[R35A]. The results showed an approximately twofold increase in the interaction between C/EBPα[R35A] and PU.1 compared to C/EBPα[WT] and PU.1 (*Figure 3C*). Finally, performing a proximity ligation assay (PLA) to study the protein's interaction in live cells showed a significantly higher number of fluorescent nuclear dots for the combination of C/EBPα[R35A] with PU.1 than with C/EBPα[WT] (*Figure 3D*), again indicating a higher affinity between the two proteins.

In sum, motif analyses showed an enrichment of the PU.1 motif at both B- and M-GREs that were differentially accessed by the mutant. Moreover, the results obtained by Co-IP, mass spectrometry, and PLA indicate that C/EBPα[R35A] shows a 1.5- to 3.5-fold higher affinity for PU.1 compared to the wild-type.

## C/EBPα[R35A] shows an increased synergy with PU.1 for myeloid gene activation in fibroblasts

Previously, we showed that the combination between C/EBPα and PU.1 is required for converting NIH 3T3 fibroblasts into macrophage-like cells (*Feng et al., 2008*). To investigate if the synergy between the C/EBPα mutant and PU.1 is altered, we isolated NIH 3T3 cell lines expressing C/EBPα[WT]-ER or C/EBPα[R35A]-ER. These were subsequently infected or not with a PU.1 retroviral construct, and Mac-1 levels monitored at different times after E2 treatment. The results showed that in the presence of PU.1, C/EBPα[R35A] activated Mac-1 more strongly than C/EBPα[WT], while cells infected with PU.1 only

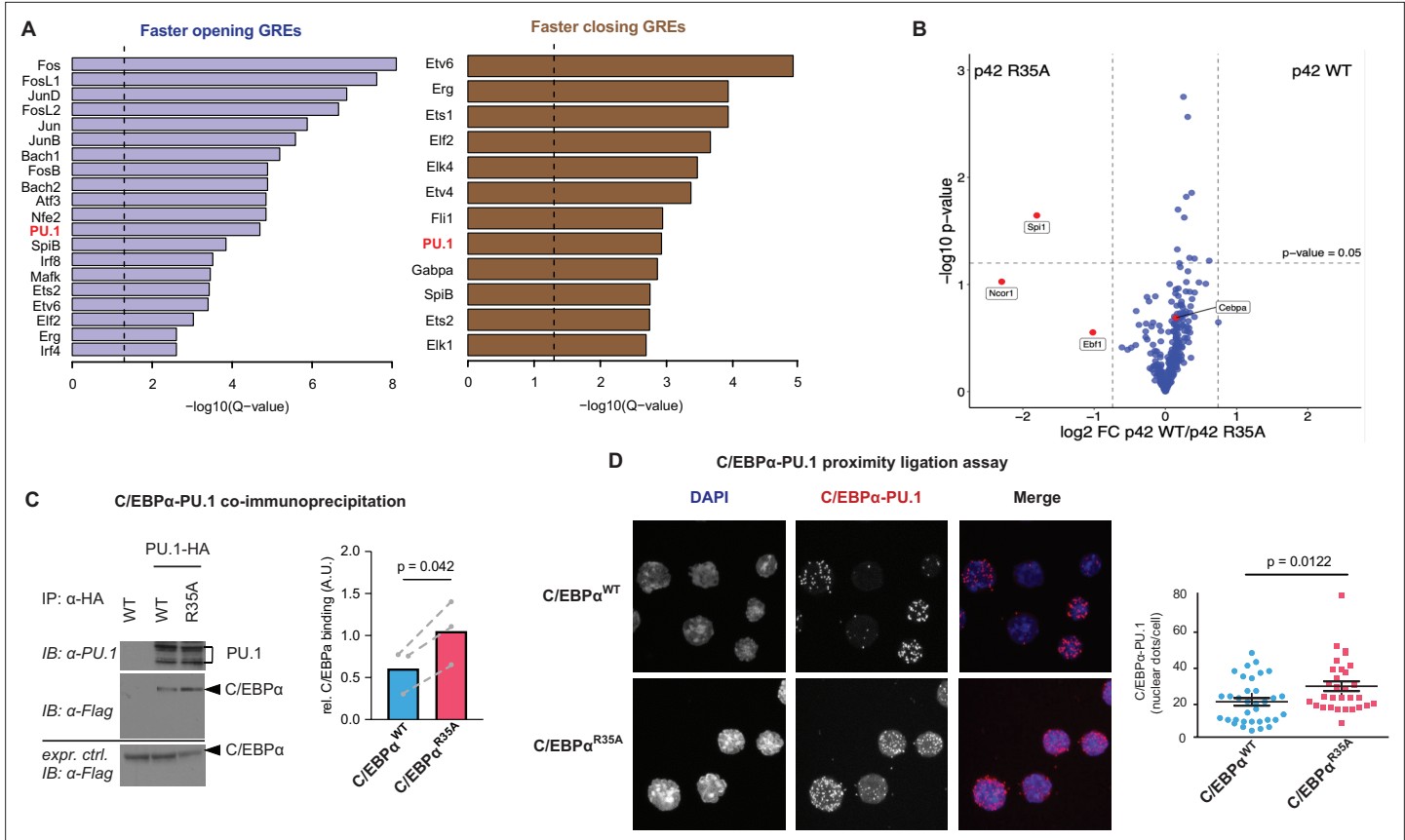

**Figure 3.** Enrichment of the PU.1 motif at differentially accessible chromatin sites and increased affinity of C/EBPα$^{R35A}$ for PU.1. (**A**) Top de novo motifs in faster opening or closing GREs induced by C/EBPα$^{R35A}$, with the PU.1 motif indicated in red. Dashed lines correspond to the significance threshold of q-value (≤0.05). (**B**) Protein interaction assay by mass spectrometry. Briefly, immortalized pre-B cells were either infected with retroviral vectors encoding doxycycline (Dox)-inducible full length (**P42**) C/EBPα$^{WT}$ or C/EBPα$^{R35A}$ proteins fused to the biotin ligase TurboID or with a TurboID only as a control. Six hours after Dox induction cells were treated with 500 μM biotin for 30 min, biotinylated proteins precipitated and analyzed by LC-mass spectrometry. Specific interactions were determined by plotting the ratio of p42 C/EBPα$^{WT}$ and C/EBPα$^{R35A}$ protein (log2-fold change) against the p-value (-log10). Vertical lines demarcate the mean ± 3 s.d. of log2-fold change ratios. Values above the horizontal line indicate significant interactions. (**C**) Western blot showing co-immunoprecipitation of PU.1 and C/EBPα in 293T cells transfected with either C/EBPα$^{WT}$ or C/EBPα$^{R35A}$ (left) and quantification of interaction experiments (n=3, right). Values were normalized for the expression of C/EBPα (mean + individual values). Dashed lines indicate paired values; statistical significance was determined using a paired Student's t-test. (**D**) Proximity ligation assay of C/EBPα and PU.1 in primary B cells induced with either C/EBPα$^{WT}$ or C/EBPα$^{R35A}$ for 24 hr. Graph shows nuclear dots per cell (mean ± s.e., n=30–34; statistical significance determined using an unpaired Student's t-test).

The online version of this article includes the following source data and figure supplement(s) for figure 3:

**Source data 1.** Electrophoretic mobility shift assay (EMSA) with 293T cells transfected with either C/EBPα$^{WT}$ or C/EBPα$^{R35A}$ incubated with a fluorophore-labeled oligonucleotide containing a palindromic C/EBPα-binding site.

**Source data 2.** Table listing 300 proteins that represent the interactome of C/EBPα showing log2-based differences in biotin labeling with the full length (**p42**) C/EBPa$^{R35A}$ protein and the p42 C/EBPα$^{WT}$ protein as well as p-values for the differences in the corresponding interactions.

**Source data 3.** Co-immunoprecipitation of PU.1 and C/EBPα in 293T cells transfected with either C/EBPα$^{WT}$ or C/EBPα$^{R35A}$ and quantification of interaction.

**Figure supplement 1.** C/EBPα DNA binding capacity and motif analyses.

(**A**) Electrophoretic mobility shift assay of nuclear extracts from 293T cells transfected with either C/EBPα$^{WT}$ or C/EBPα$^{R35A}$ and incubated with a fluorophore-labeled oligonucleotide containing a palindromic C/EBPα-binding motif (left). Protein expression control of nuclear C/EBPα proteins by western blot (middle) and densitogram-based relative DNA binding versus protein expression (right). (**B**) PU.1 enriched motifs with consensus sequence in the + and – strand shown on top, as well as several additional matched enriched de novo motifs. (**D**) De novo motifs matched to known transcription factor (TF) motifs in transiently opened GREs. The top 15 motifs are shown in the order of their significance (log10 p-values).

showed weak Mac-1 activation and C/EBPα only cells remained negative (*Figure 4A*, *Figure 4—figure supplement 1A*). Additionally, cells co-expressing C/EBPα$^{R35A}$ and PU.1 displayed more striking morphological changes than those with C/EBPα$^{WT}$ and PU.1 (*Figure 4—figure supplement 1B*). Cycloheximide treatment experiments showed that C/EBPα$^{R35A}$ exhibits a similar stability than wild-type protein and under steady-state condition is expressed at 20–30% lower levels (*Figure 4—figure supplement 1C*). This indicates that the observed gain of function of the mutant cannot simply be explained by an increased stability or expression.

## Single-molecule tracking experiments reveal that C/EBPα$^{R35A}$ in the presence of PU.1 dramatically enhances chromatin opening in fibroblasts

To investigate the effect of C/EBPα$^{WT}$ and C/EBPα$^{R35A}$ on chromatin structure and the influence of PU.1, we performed single-molecule tracking (SMT) experiments. SMT allows to observe the movements of individual TF molecules in living cells (*Liu and Tjian, 2018*) and has been recently used to quantify the interactions of TFs with open versus closed chromatin, by scaling to the mobility of single nucleosomes (*Lerner et al., 2020*; https://www.biorxiv.org/content/10.1101/2022.12.22.521655v1). For this purpose, we generated NIH3T3 cell derivatives that stably express different Halo-tagged proteins: histone H2B, Dox-inducible C/EBPα$^{WT}$, and C/EBPα$^{R35A}$. Measuring the radius of confinement and average displacements (https://star-protocols.cell.com/protocols/324) in cells expressing histone H2B-Halo served to define low versus high mobility chromatin (LoMC and HiMC), corresponding to compact versus open states, respectively (*Lerner et al., 2020*; *Figure 4C*). C/EBPα$^{WT}$-Halo and C/EBPα$^{R35A}$-Halo cultures were split and either infected with a PU.1 lentiviral construct or left uninfected (*Figure 4D*). Cells were then treated with Dox for 6 or 24 hr to induce C/EBPα, and SMT performed on ~50 cells per condition, randomly down-sampling 20,000 single-molecule motion tracks in triplicate.

Analyses of the C/EBPα$^{WT}$-Halo and C/EBPα$^{R35A}$-Halo motion tracks at 6 hpi revealed an interaction of both proteins with LoMC and HiMC, showing a moderate decrease of HiMC for C/EBPα$^{R35A}$ compared to cells induced for C/EBPα$^{WT}$ (*Figure 4E and F*). In contrast, C/EBPα$^{R35A}$-Halo cells co-expressing PU.1 showed a substantially decreased interaction with LoMC and increased interaction with HiMC, while PU.1 only moderately affected the distribution of C/EBPα$^{WT}$-Halo particles (*Figure 4E and F*). At 24 hpi, both forms of C/EBPα showed decreased interaction with LoMC and increased interaction with HiMC, although the differential effects of PU.1 co-expression were milder than at 6 hpi (*Figure 4G*).

Altogether, the SMT results obtained with NIH3T3 fibroblasts overexpressing C/EBPα and PU.1 demonstrate that PU.1 synergizes with C/EBPα in inducing substantial chromatin occupancy changes from closed to open chromatin states at 6 hpi. The synergism is much stronger for C/EBPα$^{R35A}$ than for C/EBPα$^{WT}$ and becomes less pronounced at 24 hpi, indicating that wild-type C/EBPα catches up in its ability to synergize with PU.1. These findings support the observed ability of C/EBPα and PU.1 to interact with nucleosomes in vitro and to act as a pioneer factors (*Fernandez Garcia et al., 2019*; *Lerner et al., 2020*; *van Oevelen et al., 2015*).

## C/EBPα and PU.1 binding and chromatin accessibility changes at GREs at a selected locus harboring both a B cell- and a macrophage-associated gene

The findings described so far pointed to the possibility that PU.1 acts as a coordinator of the B cell to macrophage conversion. A plausible scenario therefore is that C/EBPα removes PU.1 from B cell GREs and induces its relocation to macrophage GREs. Moreover, through its enhanced interaction with PU.1, C/EBPα$^{R35A}$ might be more efficient at relocating the factor, explaining the observed acceleration of BMT. To test this hypothesis, we conducted DNA binding experiments by chromatin immunoprecipitation followed by high-throughput sequencing (ChIP-seq) for PU.1 and C/EBPα with primary B cells expressing C/EBPα$^{WT}$ER and C/EBPα$^{R35A}$ER 3 hr after E2 induction. The data obtained were then visualized on the UCSC browser in parallel with our ATAC-seq data with cells from 0, 1, 3, 6, 18, and 120 hpi. The main messages of our results are encapsulated by the *Cyfip2-Havcr2* locus, a 270 Kb genomic region containing both a B cell- and a macrophage-restricted gene (*Figure 5A*).

Of the five genes contained in this locus, two, namely *Cyfip2* and *Havcr2,* become down- and upregulated, respectively, during BMT, with C/EBPα$^{R35A}$ inducing an acceleration of both processes

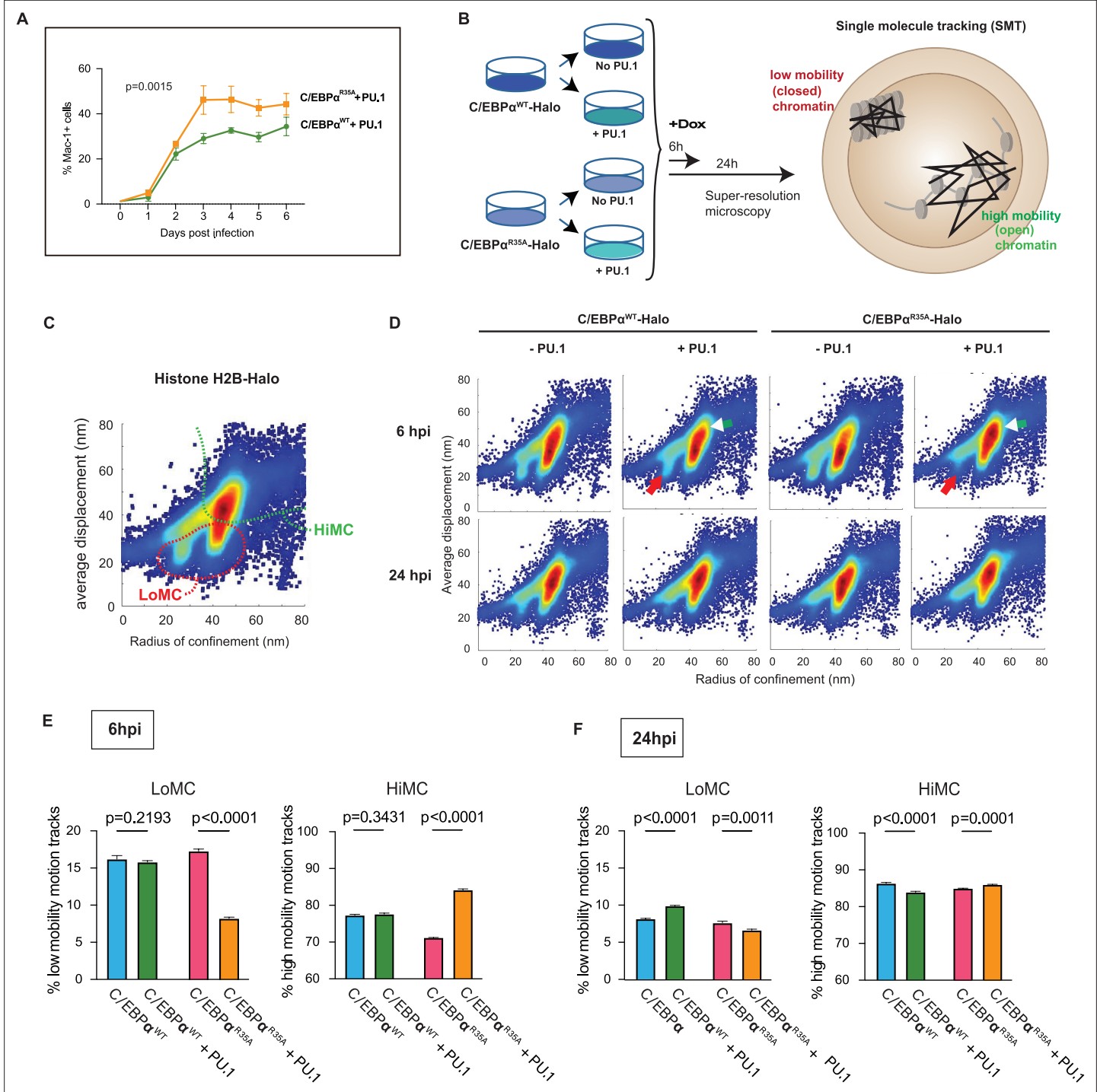

**Figure 4.** Synergy of PU.1 with C/EBPα^WT and C/EBPα^R35A in activating myeloid gene expression and chromatin accessibility changes in fibroblasts. (**A**) NIH3T3 fibroblasts expressing C/EBPα^WT-ER or C/EBPα^R35A-ER were transfected with a PU.1 construct for 24 hr and induced by E2 for the indicated times, followed by FACS to monitor Mac-1 expression (mean ± s.d., n=3; statistical significance was determined using two-way ANOVA). (**B**) Strategy used for the single-molecule tracking (SMT) experiments: NIH 3T3 cells stably expressing doxycycline (Dox)-inducible C/EBPα^WT-Halo or C/EBPα^R35A-Halo (3T3aER-R and 3T3aER-A cells) were split and transfected or not with PU.1 for 24 hr and induced with Dox for either 6 or 24 hr. They were then used for SMT by HILO super resolution microscopy, allowing to determine the radius of confinement and average displacement parameters. Particles with a small radius of confinement and average displacement correspond to low mobility chromatin (LoMC); particles with a large radius of confinement and average displacement correspond to high mobility chromatin (HiMC). (**C**) Image shows two-parameter H2B-Halo distribution in chromatin of NIH-3T3 cells, used for calibration purposes, with a stippled red line indicating area of LoMC and a green line HiMC. (**D**) Two-parameter particle distribution of 3T3aER-R and 3T3aER-A cells at 6 hpi, with red arrows pointing to differences in LoMC between the various conditions and white arrowheads

*Figure 4 continued on next page*

*Figure 4 continued*

to differences in HiMC. (**E**) Quantification of two-parameter single-cell motion tracks at 6 hpi. Values represent mean ± s.d., obtained from 20,000 randomized and down-sampled motion tracks per cell (n=3); statistical significance was determined using two-way ANOVA with multiple comparisons. (**F**) Same as (**E**) but cells induced for 24 hr.

The online version of this article includes the following figure supplement(s) for figure 4:

**Figure supplement 1.** C/EBPα^R35A hastens the relocation of PU.1 from B cell to myeloid GREs.

(*Figure 5B*). Mining of published data (*Mullen et al., 2011*) showed that PU.1 binds to four sites 50–150 kb upstream of *Cyfip2* in B cells but is essentially absent at these sites in macrophages (highlighted in red). Conversely, both PU.1 and C/EBPα bind to the promoter of *Havcr2* and three additional GREs in macrophages but not in B cells (highlighted in green, *Figure 5C*), showing that PU.1 can bind to both B- and M-GREs while C/EBPα can only bind to M-GREs.

Analyzing the PU.1 binding sites in primary B cells induced by C/EBPα for 3 hr, the factor showed higher occupancy at the B-GREs of C/EBPα^WT cells and a modestly increased binding at the M-GREs of C/EBPα^R35A cells (*Figure 5D*). Similarly, C/EBPα^WT strongly bound to the B-GREs and weakly to the M-GREs, while C/EBPα^R35A was selectively bound at the M-GREs (*Figure 5E*, *Figure 5—figure supplement 1A*). Since the DNA binding data only represent a snapshot in time, we also analyzed our dynamic ATAC-seq data for the *Cyfip2-Havcr2* locus and found that the mutant accelerates both the closure of B-GREs and opening of M-GREs (*Figure 5—figure supplement 1B*). Notably, while C/EBPα^WT triggered a transient and partial opening of B-GREs at 1 hpi and closure at 18 hpi, the mutant induced an immediate and progressive closure (*Figure 5F*). Similar effects were also seen at GREs of B cell signature genes (*Figure 2—figure supplement 1D*).

Our findings at the *Cyfip2-Havcr2* locus revealed that at 3 hpi both PU.1 and C/EBPα show distinct DNA occupancies at B- and M-GREs in wild-type and mutant expressing cells. These differences can best be explained by the hypothesis that C/EBPα triggers the redistribution of PU.1 bound to B-GREs to M-GREs.

## Genome-wide analysis of TF binding and chromatin accessibility kinetics

The finding that the PU.1 redistribution appears to be accelerated in the mutant cells predicts that B cell enhancers show lower levels of PU.1 binding in C/EBPα^R35A-induced cells than in C/EBPα^WT-induced cells. Conversely, myeloid enhancers should show higher PU.1 occupancy. To test this, we conducted a genome-wide analysis of the ChIP-seq data, revealing 4310 regions differentially bound by PU.1 at 3 hpi. These regions were divided into sites bound either *Less* or *More* in C/EBPα^R35A cells relative to C/EBPα^WT cells. As seen from the ATAC-seq data, the majority of the observed 1647 sites that were bound *Less* by PU.1 became inaccessible at 18 hpi in C/EBPα^WT cells, showing a transient opening between 1 and 6 hpi. In contrast, in C/EBPα^R35A cells these sites became already partially closed at 1 hpi and fully inaccessible after 18 hpi (*Figure 6A*). The opposite was observed for the 2650 sites bound *More* by PU.1, where chromatin accessibility increased already at 1 hpi in the mutant cells, while in wild-type cells they only opened at 120 hpi (*Figure 6D and E*). As predicted, genes associated to sites bound *Less* by PU.1 showed a strong enrichment of B cell-related GO terms while *More*-bound sites were enriched for myeloid-related GO terms (*Figure 6C*).

The redistribution model further predicts that C/EBPα^R35A is released from B cell enhancers already at 3 hpi and is instead enriched at myeloid GREs, while C/EBPα^WT shows a relocation delay. We therefore compared the genome-wide DNA binding profiles of wild-type and mutant C/EBPα proteins. Of 7144 differentially occupied sites, 4352 sites were bound *Less* by C/EBPα^R35A. Approximately 70% of these started closing at 1 hpi in C/EBPα^R35A cells while in C/EBPα^WT cells they only closed at 120 hpi, after transiently opening between 1 and 6 hpi (*Figure 6—figure supplement 1A*). The remaining 30% of the sites showed a gradual but heterogeneous increase in accessibility. Finally, about half of the sites bound *More* by C/EBPα^R35A became more rapidly opened, as expected, while the other half showed a transient opening before closing in both wild-type and mutant cells (*Figure 6—figure supplement 1B*). GO analysis for the *Less* C/EBPα^R35A-bound sites showed an association with both B cell and myeloid terms while the rapidly opening *More* bound sites were associated with myeloid terms (*Figure 6—figure supplement 1C*).

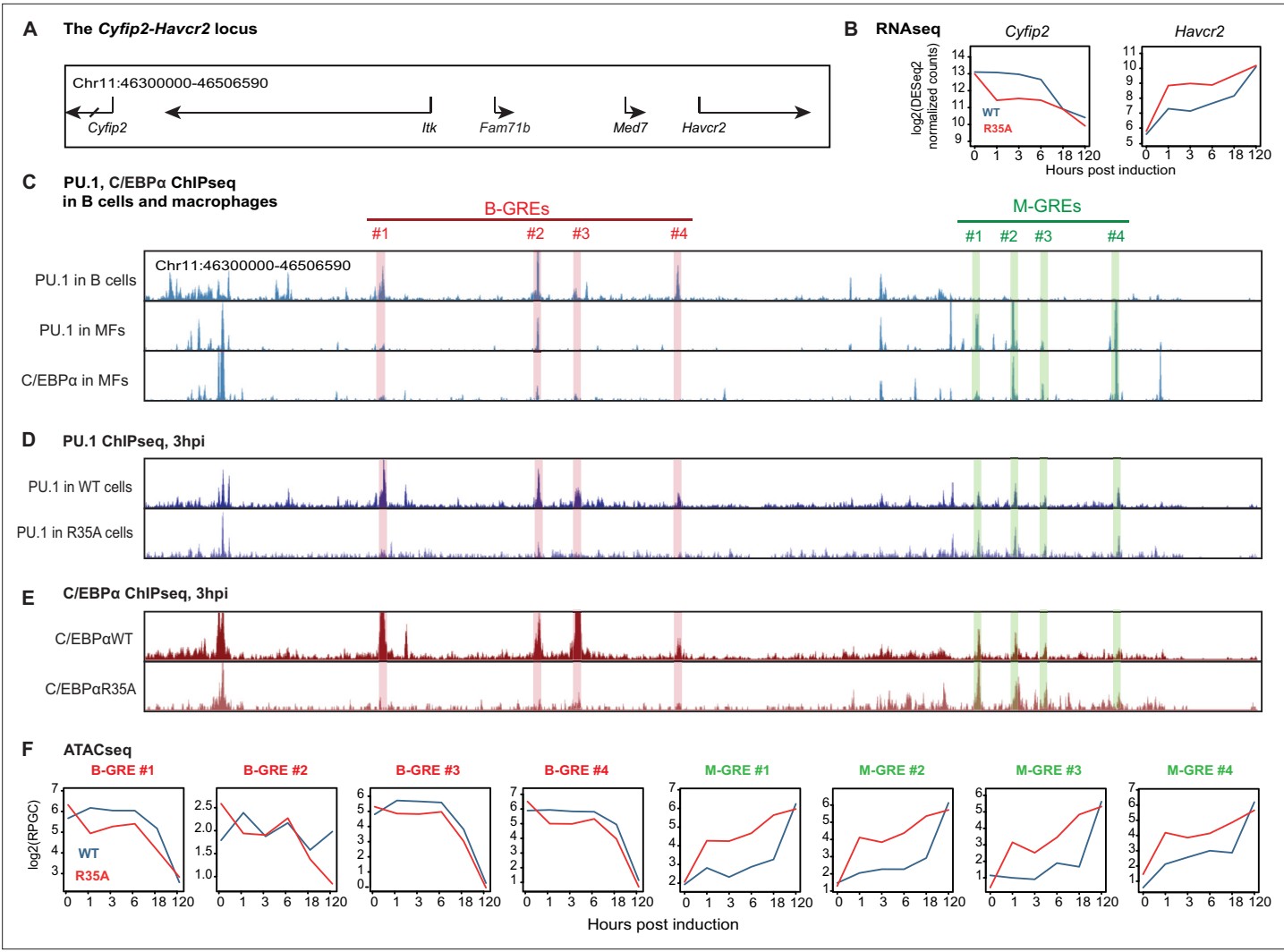

**Figure 5.** C/EBPα and PU.1 binding and chromatin accessibility changes of B cell and myeloid GREs. (A) Structure of the *Cyfip2-Havcr2* locus (chr11:46,300,000–46,506,590). (B) Kinetics of gene expression after E2 induction of C/EBPα[WT] or C/EBPα[R35A]. (C) Screenshots of the UCSC browser showing peaks corresponding to PU.1 binding in B cells and in bone marrow derived macrophages (BMDM), as well as C/EBPα binding in BMDM (gain 5000). B cell and macrophage GREs are highlighted in red and green, respectively. Coordinates of B-GREs: #1, chr11:46,343,904–46,344,618; #2, chr11:46,372,449–46,373,726; #3, chr11:46,379,961–46,380,726; #4, chr11:46,398,858–46,399,316. M-GREs: #1, chr11:46,454,489–46,455,316; #2, chr11:46,461,421–46,462,284; #3, chr11:46,466,901–46,467,255, #4, chr11:46,480,463–46,480,843. (D) Screenshots of PU.1 binding in B cells induced with E2 for 3 hr to activate expression of C/EBPα[WT] or C/EBPα[R35A] (gain 2000). (E) As in (D), but for C/EBPα binding. (F) Kinetics of chromatin accessibility, determined by assay for transposase-accessible chromatin sequencing (ATAC-seq) of B-GREs (framed in red) and M-GREs (in green) in C/EBPα[WT] or C/EBPα[R35A]-induced cells. The chromatin immunoprecipitation followed by high-throughput sequencing (ChIP-seq) data represent RPKM normalized values, the ATAC-seq data represent RPGC normalized values.

The online version of this article includes the following figure supplement(s) for figure 5:

**Figure supplement 1.** DNA binding of PU.1 and C/EBPa and assay for transposase-accessible chromatin sequencing (ATAC-seq) kinetics during B cell to macrophage transdifferentiation (BMT).

Finally, we tested sites bound *Less* by both PU.1 and C/EBPα[R35A] and found 783 shared sites, whereas 385 sites were bound *More* by both PU.1 and mutant C/EBPα. In contrast, no overlap was found between PU.1 bound *Less* and C/EBPα bound *More* or vice versa (**Figure 6—figure supplement 1D**) showing a strong positive correlation in the binding of the two factors. The chromatin accessibility dynamics at these commonly bound regions reflected by and large the observations described for the singly bound factors and showed the expected association with B cell and myeloid-associated GO terms (data not shown).

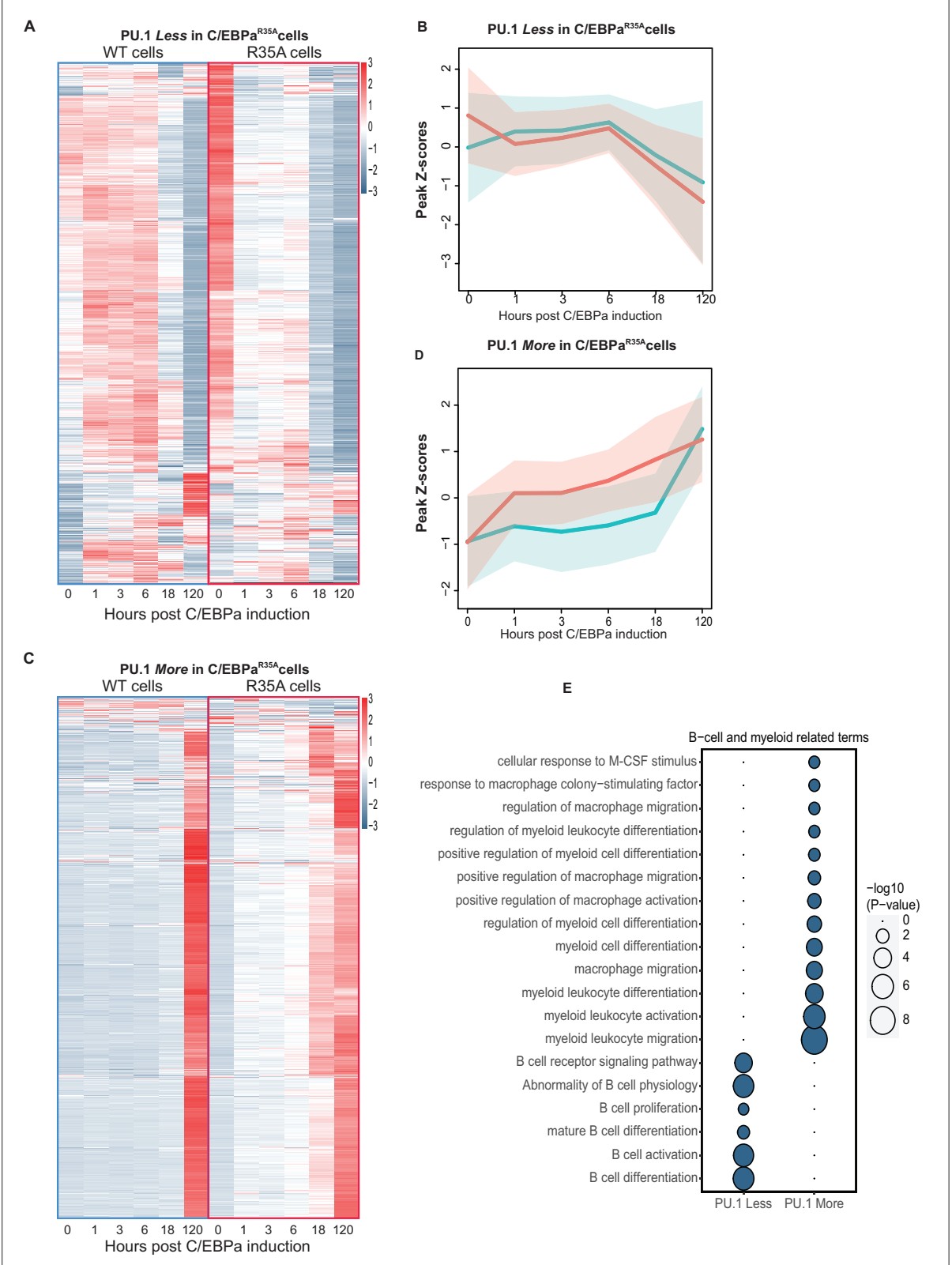

**Figure 6.** Analysis of chromatin accessibility changes during B cell to macrophage transdifferentiation (BMT) at GREs differentially bound by PU.1 in C/EBPα$^{WT}$ and C/EBPα$^{R35A}$-induced B cells. (**A**) Heat maps of ATAC peaks at sites bound *Less* by PU.1 in C/EBPα$^{R35A}$ cells compared to C/EBPα$^{WT}$ cells and line diagrams showing a comparison of the kinetics. Blue and red lines represent median values (log2) for wild-type and mutant C/EBPα, respectively,

*Figure 6 continued on next page*

*Figure 6 continued*

with shadows representing the 95% confidence interval. (**C** and **D**) Same as for (**A** and **B**), but for sites bound *More* by PU.1. (**E** and **F**) Gene Ontology analysis for B cell and myeloid-related terms, with size of circles being proportional to their p-values (log10).

The online version of this article includes the following figure supplement(s) for figure 6:

**Figure supplement 1.** Analysis of chromatin accessibility changes during B cell to macrophage transdifferentiation (BMT) at regions differentially bound by C/EBPα in 3 hr induced C/EBPα$^{WT}$ and C/EBPα$^{R35A}$ cells.

In conclusion, our genome-wide analyses showed that C/EBPα induces the relocation of PU.1 from B- to M-GREs, in a process accelerated by the mutant. We also observed that C/EBPα$^{WT}$ transiently opens B-GREs during B cell silencing, while C/EBPα$^{R35A}$ induces their immediate closing. The finding that the chromatin of B-GREs is more accessible in uninduced C/EBPα$^{R35A}$ cells suggests a leakiness of the construct and could at least in part explain this finding. However, this poised state does not result in an altered expression of associated B cell genes compared to uninduced C/EBPα$^{WT}$ cells. Finally, our data showed the progressive closing/opening of B/M-GREs between 1 and 6 hpi, while gene expression during this phase remains essentially unchanged (*Figure 1—figure supplement 1F, G*; *Figure 2—figure supplement 1C, D*; *Figure 5—figure supplement 1B*). These observations suggest that chromatin restructuring can be uncoupled from changes in gene regulation.

## Carm1 asymmetrically di-methylates arginine 35, decreasing the affinity of C/EBPα for PU.1

The observed accelerated BMT induced by C/EBPα with an alanine replacement of arginine 35 could either be due to the replacement of a charged amino acid with a hydrophobic residue or to the loss of arginine methylation, or both. To test this, we replaced the arginine with the charged residue lysine in our inducible retroviral vector and expressed it in primary B cells. These cells were then induced with E2 for 1–120 hr and analyzed by RNA-seq and ATAC-seq in comparison to C/EBPα$^{WT}$-induced cells. PCA of the data showed an accelerated BMT time course of gene expression and chromatin accessibility (ATAC peaks) for the C/EBPα$^{R35K}$-induced cells (*Figure 7—figure supplement 1A,B*), similar but not identical to those seen with C/EBPα$^{R35A}$ cells (not shown). This suggests that the C/EBPα$^{R35A}$-induced BMT acceleration is not simply due to a lack of charge at the R35 residue.

To explore whether R35 is asymmetrically di-methylated, we used a previously described human B cell to macrophage BMT system (*Rapino et al., 2013*). For this we generated 4-hydroxytamoxifen (4-OHT)-inducible lines expressing C/EBPα$^{WT}$ and C/EBPα$^{R35A}$ fused to ERT-2, called BLaER2 and BLaER2-A, respectively. After treating the cells with 4-OHT for 24 hr and immunoprecipitating C/EBPα followed by western blotting and staining of the blot with an antibody specific for asymmetrically di-methylated arginine-containing proteins, C/EBPα could be detected in BLaER2 but not in BlaER2-A cells, as expected (*Figure 7A*). Similar results were obtained with transfected 293T cells, showing methylated C/EBPα in the wild-type but not mutant cells (*Figure 7B*).

To identify which of the Prmts known to asymmetrically methylate arginine is responsible, 293T cells were co-transfected with C/EBPα$^{WT}$ or C/EBPα$^{R35A}$ and with either Prmt1, Prmt3, Carm1 (Prmt4), or Prmt6. Only Carm1 induced asymmetric di-methylation of C/EBPα (as detected with an antibody specific for the modification) and that this was restricted to the wild-type protein (*Figure 7—figure supplement 2A*). These data indicate that Carm1 specifically and asymmetrically di-methylates R35.

We next performed Co-IP experiments to examine whether C/EBPα and Carm1 can interact, using 293T cells co-transfected with the two proteins, and found that both C/EBPα$^{WT}$ and C/EBPα$^{R35A}$ are capable of interacting with Carm1 (*Figure 7C*). Next, we performed a PLA with NIH 3T3 cell derivatives expressing either C/EBPα$^{WT}$-ER or C/EBPα$^{R35A}$-ER. After 24 hr of induction with E2, cells were stained with antibodies to C/EBPα and Carm1 and a secondary antibody to detect their interaction. This again demonstrated that both forms of C/EBPα can interact with Carm1, with C/EBPα$^{R35A}$ cells showing a slightly higher signal (*Figure 7—figure supplement 2B*).

To determine which of the 20 arginines in C/EBPα can be bound by Carm1, we conducted an in vitro methylation assay using 14 synthetic peptides spanning the entire molecule. Only the peptide containing arginine 35 generated a signal (*Figure 7—figure supplement 2C*). Next, we studied the enzyme's specificity for selected targets. For this we chose as substrates a 40 amino acid long C/EBPα peptide with either an unmethylated R35, an asymmetrically di-methylated R35, or an alanine

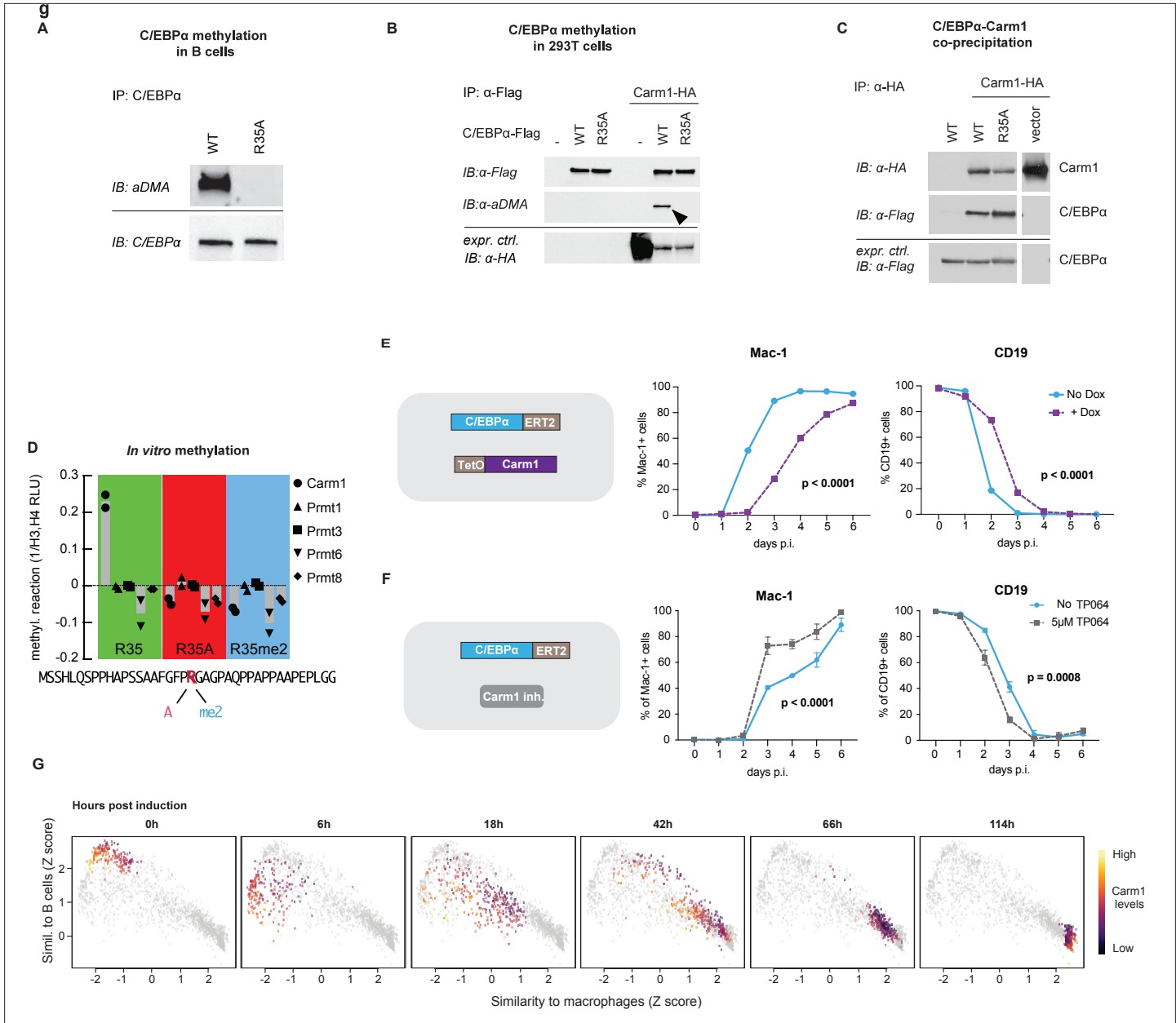

**Figure 7.** Carm1 asymmetrically di-methylates arginine 35 and the enzyme's level correlates with the speed of C/EBPα-induced B cell to macrophage transdifferentiation (BMT). (**A**) Immunoprecipitation (IP) and immunoblotting (IB) of C/EBPα and asymmetrically di-methylated arginine (aDMA) containing proteins. (**B**) Immunoprecipitation of C/EBPα from 293T cells co-transfected with either C/EBPα^WT-Flag or C/EBPα^R35A-Flag with or without Carm1-HA, followed by immunoblot with an antibody against aDMA, Flag, and HA. (**C**) As in (**B**), but immunoprecipitation of Carm1 from co-transfected 293T cells and immunoblotted with antibodies against Flag and HA. (**D**) In vitro methylation assay with recombinant Carm1, Prmt1, Prmt3, Prmt6, or Prmt8 proteins together with a C/EBPα peptide (aa 15–54) containing either an unmethylated arginine 35 (R in red), an alanine replacement (A in pink), or an asymmetric di-methylation (me2 in blue). (**E**) Effect of Carm1 overexpression on BMT kinetics of human B cells measured by Mac-1 and CD19 expression (mean ± s.d., n=3, statistical significance determined using two-way ANOVA). (**F**) Same as (**E**), but showing the effect of 5 µM of TP064. (**F**) Correlation with B cell and macrophage states of Carm1 expression levels in single-cell BMT trajectories. Data extracted from previously published work (*Francesconi et al., 2019*).

The online version of this article includes the following source data and figure supplement(s) for figure 7:

**Source data 1.** Immunoprecipitation (IP) and immunoblotting of C/EBPα and asymmetrically di-methylated arginine (aDMA) containing proteins.

**Source data 2.** Immunoprecipitation (IP) of C/EBPα from 293T cells co-transfected with either C/EBPα^WT-Flag or C/EBPα^R35A-Flag with or without Carm1-HA, followed by immunoblot (IB) with antibodies (Abs) against asymmetrically di-methylated arginine (aDMA), Flag, and HA.

**Source data 3.** Immunoprecipitation (IP) of Carm1 from 293T cells co-transfected with either C/EBPα^WT-Flag or C/EBPα^R35A-Flag with or without Carm1-

*Figure 7 continued*

HA, followed by immunoblot with antibodies against Flag and HA.

**Figure supplement 1.** C/EBPα[R35K]-induced B cell to macrophage transdifferentiation (BMT): principal component analysis (PCA) of gene expression and chromatin accessibility.

**Figure supplement 1—source data 1.** Immunoprecipitation (IP) of cells co-transfected with either C/EBPα[WT]-Flag or C/EBPα[R35A]-Flag and different type I Prmts followed by immunoblot with antibodies against asymmetrically di-methylated arginine.

**Figure supplement 1—source data 2.** Western blot of Carm1 and GAPDH in B cell lines RRC3 and RAC1 with or without addition of doxycycline (Dox).

**Figure supplement 1—source data 3.** Western blot of asymmetrically di-methylated BAF155 (AsDM-BAF155) and total BAF155 (BAF155) in B cells treated with different concentrations of TP064.

**Figure supplement 2.** Protein methylation and proximity ligation assays (PLA); C/EBPα[R35A] interaction with Carm1 and effects of Carm1 perturbations on C/EBPα[R35A]-induced B cell to macrophage transdifferentiation (BMT).

replacement. The three peptides were then treated with purified Carm1, Prmt1, Prmt3, Prmt6, or Prmt8 proteins. Only unmethylated R35 peptide could be methylated Carm1, with all other conditions being negative (*Figure 7D*). To examine if methylation affects the affinity of an R35 containing C/EBPα peptide for PU.1, we analyzed the data from an interactome screen (PRISMA) (*Ramberger et al., 2021a*). The comparison of a 15 mer with either an unmethylated or an asymmetrically di-methylated R35 showed that PU.1 interacts more strongly with the unmethylated peptide (*Figure 7—figure supplement 2F*).

In summary, our results indicate that Carm1, but none of several other Prmts tested, induces the asymmetric di-methylation of specifically arginine 35 of C/EBPα and that an unmethylated peptide at R35 has a higher affinity for PU.1 than its methylated form.

## Carm1 gain- and loss-of-function experiments modulate the speed of C/EBPα-induced BMT

To evaluate the effect of Carm1 activity on BMT, we conducted gain- and loss-of-function experiments. For this, we generated two BLaER2 cell derivatives (RRC3 and RAC-1) containing 4-OHT-inducible C/EBPα[WT] and C/EBPα[R35A], respectively, and a Dox-inducible Carm1 construct. For the gain-of-function experiments we treated 4-OHT-induced cells with Dox and observed a strong Carm1 band by western blot (*Figure 7—figure supplement 2D*) while treatment with TP064 (*Nakayama et al., 2018*) reduced the level of asymmetrically di-methylated BAF155 (*Figure 7—figure supplement 2E*), a well-described target of the enzyme (*Wang et al., 2014*). Carm1 overexpression resulted in a delay of both Mac-1 activation and Cd19 silencing in 4-OHT-induced wild-type but not mutant C/EBPα-containing cells (*Figure 7F* and *Figure 7—figure supplement 2G*). Conversely, treatment of 4-OHT-induced C/EBPα[WT]-expressing cells with TP064 resulted in a robust acceleration of both Mac-1 upregulation and Cd19 silencing while the kinetics of C/EBPα[R35A]-expressing RAC1 cells were not altered (*Figure 7G* and *Figure 7—figure supplement 2H*).

To determine if endogenous Carm1 levels correlate with BMT velocity, we analyzed a single-cell gene expression dataset of primary B cells undergoing BMT, generated in a previous study (*Francesconi et al., 2019*). This showed that cells with the lowest Carm1 levels transitioned to a macrophage-like identity faster than those with higher levels, with the most substantial differences occurring during the early timepoints (*Figure 7H*), reflecting the more pronounced acceleration of gene expression within the first hours after induction of C/EBPα[R35A] (*Figure 1F*).

Our data showed that Carm1 overexpression slows down BMT, whereas Carm1 inhibition accelerates it, and that such perturbations did not affect C/EBPα[R35A]- induced BM. This is consistent with the notion that the proportion of C/EBPα with methylated versus unmethylated arginine 35 determines the velocity of BMT.

## Carm1 modulates the directionality of GMP cell differentiation

To examine if Carm1 plays a role in myeloid cell fate specification, we first analyzed its role during myeloid differentiation. During this process, common myeloid progenitors (CMPs) differentiate into granulocyte-macrophage progenitor (GMPs), which further differentiate into granulocytes and monocyte/macrophages (*Figure 8A*), requiring C/EBPα (*Zhang et al., 1997*; *Zhang et al., 2004*; *Ohlsson et al., 2016*).

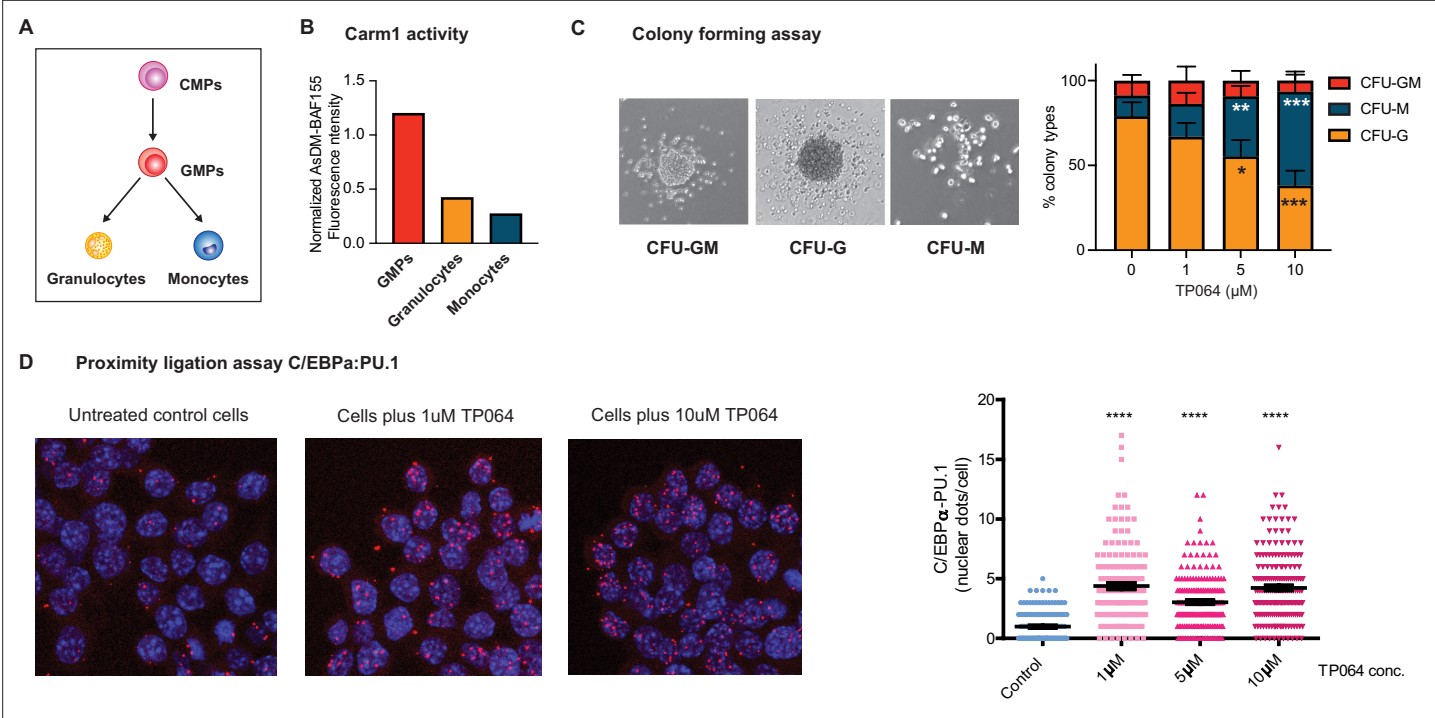

**Figure 8.** Effect of Carm1 activity on myeloid differentiation and C/EBPα-PU.1 interaction. (**A**) Simplified representation of myeloid differentiation. Common myeloid progenitors (CMPs); granulocyte-macrophage progenitors (GMPs). (**B**) Levels of AsDM-BAF155 relative to total BAF155 in GMPs, granulocytes and monocytes as a proxy for Carm1 activity. (**C**) On the left, representative images of colony types obtained from GMPs grown in Methocult. On the right, quantification of colony numbers obtained in cultures without or with various concentrations of the Carm1 inhibitor TP064 for 14 days, showing percentage of bipotent (CFU-GM), monocytic (CFU-M) and granulocytic (CFU-G) colonies (mean ± s.d., n=3–4), statistical significance was determined using a one-way ANOVA for each cell type. (**D**) Proximity ligation assay of endogenous C/EBPα and PU.1 in the mouse macrophage cell line RAW 264.7 treated for 24 hr with TP064 or left untreated. On the left confocal microscopy images. On the right, counts of nuclear dots per cell (mean ± s.e., n=149–190 cells per condition). Four stars: p<0.0001 (statistical significance determined using an unpaired Student's t-test.).

The online version of this article includes the following figure supplement(s) for figure 8:

**Figure supplement 1.** Carm1 activity in myeloid cells, effect of TP064 on viability of colony forming cells and expression of C/EBPα and PU.1 in cells used for proximity ligation assay (PLA).

Examining the expression of Carm1 in CMPs, GMPs, granulocytes, and macrophages (*Choi et al., 2019*) revealed a steady decrease during differentiation (*Figure 8—figure supplement 1A*). We next sorted GMPs, granulocytes, and monocytes from the bone marrow and measured AsDM-BAF155 as a proxy for Carm1 activity (*Wang et al., 2014*) relative to total BAF155. The highest Carm1 activity was detected in GMPs, with a 3.5-fold decrease in granulocytes and a further 4.5-fold decrease in monocytes (*Figure 8B* and *Figure 8—figure supplement 1B*). This showed that not only the expression of Carm1 but also its activity decreases during myeloid differentiation.

To investigate whether Carm1 activity influences the direction of differentiation of a bipotent precursor, we conducted a colony assay with sorted GMPs in the absence or presence of the Carm1 inhibitor TP064. For this, GMPs were seeded in semisolid medium containing 0, 1, 2.5, or 10 µM TP064 and the number of granulocytic (CFU-G), monocytic (CFU-M), and mixed (CFU-GM) colonies scored after 12 days (*Figure 8—figure supplement 1*). The results showed a significant dose-dependent reduction of granulocyte colonies and a corresponding increase in monocyte colonies (*Figure 8C*). The observed bias was not due to a granulocyte-specific toxicity of the inhibitor, as the total number of colonies remained unchanged (*Figure 8—figure supplement 1C*).

Our findings suggest that Carm1 modulates the directionality of differentiation of GMPs, with low activities increasing the proportion of monocytic colonies while reducing the proportion of granulocytic colonies.

## Carm1 inhibition increases the interaction between endogenous C/EBPα and PU.1

The finding that C/EBPα^R35A interacts more strongly with PU.1 than C/EBPα^WT could represent an artifact of overexpression. Therefore, to investigate whether the ability of endogenous C/EBPα to interact with PU.1 can be modulated by the levels or activity of Carm1, we treated the macrophage cell line RAW 264.7 for 24 hr with 0, 1, 5, or 10 μM TP064 and subjected the cells to a PLA, using antibodies against C/EBPα and PU.1. Our results showed a four- to fivefold increase in the number of nuclear dots in treated compared to untreated cells (*Figure 8D*), indicating that the inhibition of methylation of endogenous C/EBPα increased the factor's affinity for interaction with PU.1. The finding that C/EBPα and PU.1 are expressed at similar levels in treated and untreated cells indicates that the inhibitor does not alter the proteins' expression and thus could not explain the observed PLA differences (*Figure 8—figure supplement 1D*).

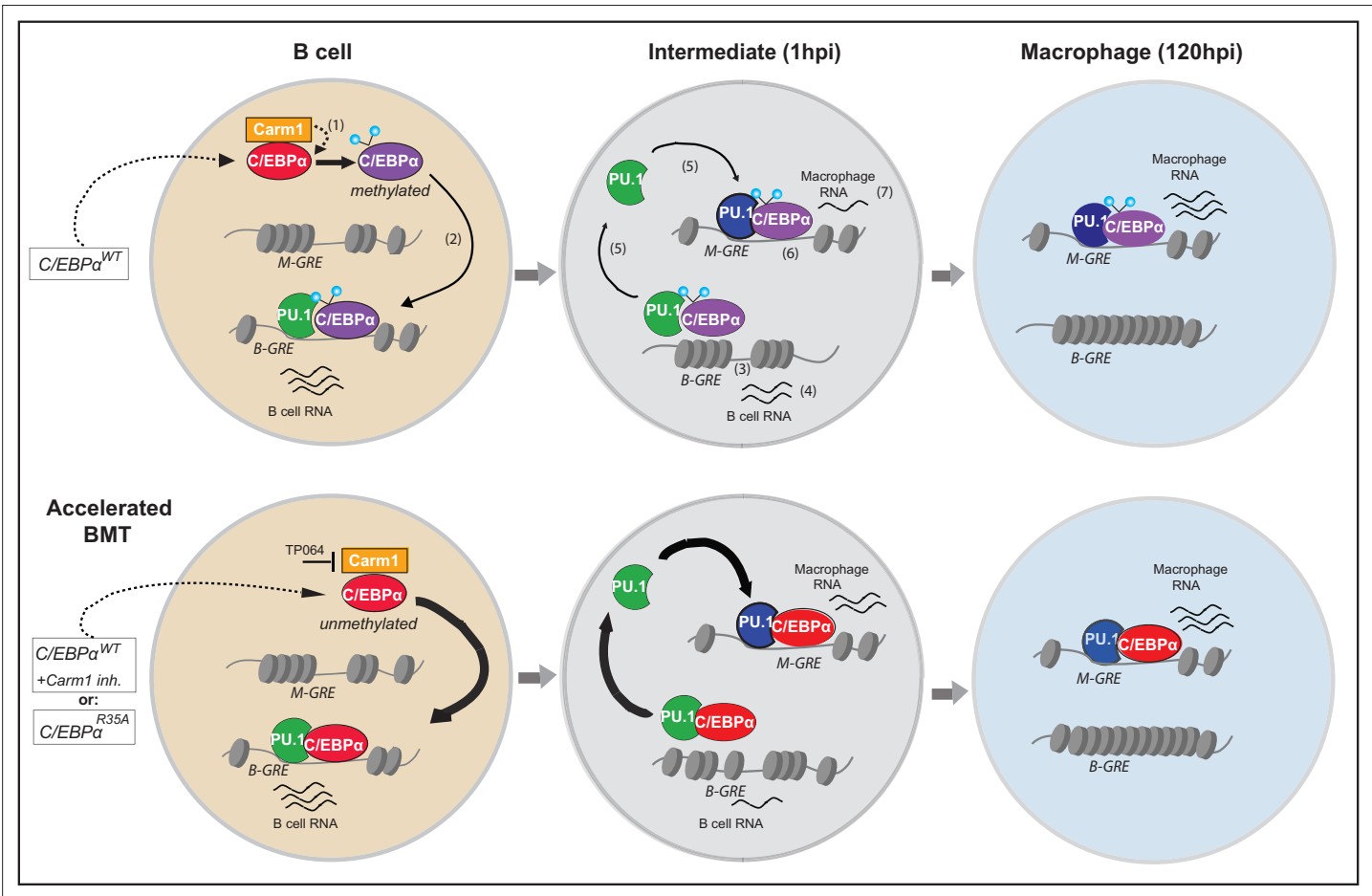

**Figure 9.** Diagram summarizing the mechanism of C/EBPα-induced B cell to macrophage transdifferentiation (BMT) and its acceleration by the R35 mutant. Starting with a B cell shortly after E2-induced translocation of C/EBPαER into the nucleus, the C/EBPα-induced BMT occurs in various steps, illustrated in the upper row. In a first step incoming C/EBPα is bound by Carm1 and becomes asymmetrically di-methylated on arginine 35 (1). Then, C/EBPα interacts with PU.1, bound to B cell-restricted gene regulatory elements (B-GREs) (2). This induces PU.1 release, chromatin closing at B-GREs (3) and downregulation of associated B cell genes (4). Free PU.1 (possibly in a complex with C/EBPα) relocates to macrophage-restricted GREs (M-GREs), which are also bound by C/EBPα (5) resulting in chromatin opening at M-GREs (6) and upregulation of associated macrophage genes (7). As shown in the lower row, unmethylated C/EBPα, either because it is mutated at R35 or Carm1 is inhibited, interacts with a higher affinity for PU.1 than the wild-type, leading to the acceleration of all subsequent steps. In the diagram PU.1 is represented as a green icon when acting as a B cell regulator and as a blue icon after being repurposed as a myeloid regulator. Methylated and unmethylated C/EBPα are represented by purple and red icons (their depiction ignores the fact that C/EBPα binds as homo- or heterodimers). In the model we assume that methylated and unmethylated C/EBPα have distinct but overlapping functions. However, it is possible that C/EBPα^WT cells contain a mixture of methylated and unmethylated C/EBPα, and that only the unmethylated form is active and can induce BMT. We assume that the methylated form is active in other cellular contexts, such as during granulocyte specification.

Our observations indicate that endogenous C/EBPα can interact with PU.1 and that this interaction is regulated by Carm1. Thus, inhibition of the enzyme results in a macrophage cell line induces an enhanced interaction between C/EBPα and PU.1, similar to the interaction of exogenous C/EBPα[R35A] with endogenous PU.1.

## Discussion

Here, we describe a mechanism by which the speed of C/EBPα-induced BMT can be regulated. It is based on the identification of a C/EBPα missense mutant in a single arginine (R35), located within the unstructured transactivation domain, and which is specifically targeted by the methyltransferase Carm1. As summarized in the diagram of *Figure 9*, an alanine replacement of R35 or Carm1 inhibition dramatically accelerates the velocity of BMT. This is based on an increased affinity between C/EBPα and its obligate partner PU.1. The C/EBPα and PU.1 combination induces a rapid closing of B cell GREs and opening of macrophage of GREs, further accelerated by the mutant. In addition, C/EBPα is capable of redistributing PU.1 from B cell to macrophage GREs, in a process again enhanced by the mutant, resulting in the coordinated silencing of B cell genes and activation of macrophage genes.

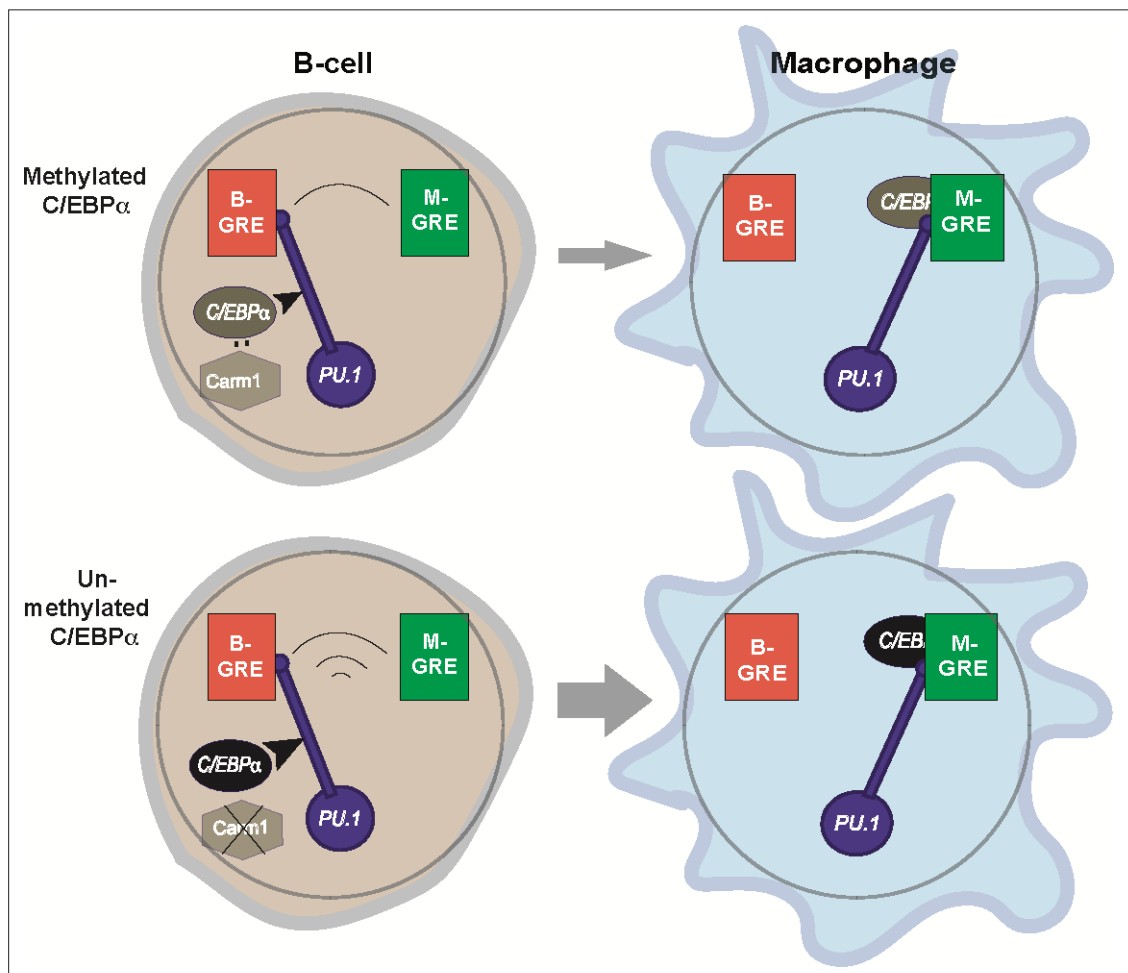

**Figure 10.** Relay model of B cell to macrophage transdifferentiation (BMT). In this model we propose that PU.1 acts as a relay that coordinates the silencing of B cell-restricted and activation of macrophage-restricted genes. The relay is triggered by C/EBPα entering into the nucleus (curved arrow) where it can interact with Carm1, which in turn can modulate C/EBPα activity (for details, see *Figure 9*). PU.1 is initially bound to B-GREs in B cells, and during BMT gets relocated to M-GREs that are co-occupied by C/EBPα. The relay velocity is tuned by the potency of incoming C/EBPα, that is, whether or not the factor is methylated. The mutually antagonistic regulation of B cell and macrophage genes enabled by this mechanism ensures the faithful generation of the macrophage phenotype and prevents the formation of stable cell states that express mixed lineage markers ('confused cells'). Of note, the relay mechanism is unidirectional since about 24hpi B cells become irreversibly committed to macrophages, due to the upregulation of endogenous *Cebpa* that maintains the novel cell state (***Bussmann et al., 2009***).

Our data indicate that the interaction affinity between C/EBPα and endogenous PU.1, already expressed in B cells, is rate limiting, as the observed higher affinity of mutant C/EBPα is sufficient to accelerate all subsequent BMT steps. In this scenario, C/EBPα acts as a trigger for PU.1, which serves as a 'relay' that couples the concomitant silencing and activation of gene expression (*Figure 10*). This mechanism avoids the formation of 'confused' cells, such as cells that upregulate the macrophage program without silencing the B cell program or vice versa, and therefore ensures the faithful establishment of the macrophage transcriptome. It will be interesting to determine whether the structure of PU.1 changes as it is repurposed from a B cell to a macrophage regulator during BMT, and if so, whether this reflects alternative forms of PU.1, such as recently described in an in vitro study (*Xhani et al., 2020*). Of note, we have not attempted to integrate into our model the fact that PU.1 is expressed at approximately twice the levels in macrophages than in B cells (*Singh et al., 1999*).

A mechanism similar to the C/EBPα-induced redistribution ('theft') of PU.1 from B- to M-GREs by an incoming TF as described here has recently been described for Gata6-induced reprogramming of ESCs into primitive endoderm (*Thompson et al., 2022*) and might also be operative during the Oct4-induced reprogramming of Sox2-expressing neural progenitors to induced pluripotent stem cells (*Kim et al., 2009*). A theft mechanism involving PU.1 has also been reported for T cell differentiation although in this case PU.1 acts as the 'thief' by redirecting the Runx1 and Satb1 to novel sites (*Hosokawa et al., 2018*). Another relevant example is the sequestering of Gata3 by Tbet from Th2 genes and its relocation to Th1 genes during Th1 cell specification (*Hertweck et al., 2022*). Thus, TF-mediated redistribution by a theft or 'relay' mechanism as proposed here may be a more widespread phenomenon governing cell fate decisions during differentiation.

Arginine side chains can also be converted by peptidylarginine deiminases to citrulline residues that cannot be methylated by Carm1 (*Cuthbert et al., 2004*) and therefore deimination at R35 might be involved in regulating C/EBPα activity. However, against this possibility is our previous finding that R35 is methylated but not citrullinated in myeloid cells (*Ramberger et al., 2021b*). Another argument is the observed capacity of TP064 to accelerate BMT, as this is a highly specific inhibitor of Carm1 that blocks the substrate binding site of the enzyme (*Nakayama et al., 2018*).

The location of arginine 35 within the IDR of C/EBPα, together with its ability to regulate the activity of the factor, raises the possibility that C/EBPαR35A generates a locally structured stretch within the IDR that acts as a substrate for PU.1. Alternatively, the IDRs of C/EBPα and PU.1 might engage in weak interactions to form transcriptional condensates *Boija et al., 2018* whose properties might differ between wild-type and mutant C/EBPα. The initial inspection of the C/EBPα DNA binding profiles at 3 hpi suggested that C/EBPα$^{WT}$ and C/EBPα$^{R35A}$ have different DNA binding preferences for B- and M-GREs. However, the observation of similar 'preferences' by PU.1 suggests that they reflect differences in the timing of PU.1:C/EBPα release from B-GREs and binding to M-GREs. The kinetics of chromatin accessibility changes over time at the same sites observed by ATAC-seq further supports this interpretation.

The findings described help explain how C/EBPα and PU.1 co-operate to generate macrophage-like cells from fibroblasts (*Feng et al., 2008*). Thus, while C/EBPα$^{WT}$ can bind to closed chromatin and induce a transition to open chromatin, the combination with PU.1 does so faster and this is even further accelerated with C/EBPα$^{R35A}$, at speeds exceeding that of other pioneer TFs studied so far (*Lerner et al., 2020*). Similar to our observations of gene expression during BMT, in this system the differences between the wild-type and the mutant are much more pronounced at an early timepoint, suggesting that the chromatin changes in C/EBPα$^{WT}$ cells eventually catch up.

Our research has uncovered a close connection between the speed of cell fate choice and directionality during forced and normal myeloid differentiation. Such a link may also have implications for the formation of cancer cells. Thus, Carm1-deficient mice are resistant to MLL-AF9-induced acute myeloid leukemia (*Greenblatt et al., 2016*; *Greenblatt et al., 2018*), probably because a critical target protein cannot be methylated by the enzyme. This target might be C/EBPα itself, which had been shown to be required for the formation of MLL-AF9-induced AML (*Ye et al., 2015*; *Ohlsson et al., 2016*). Our findings suggest that in GMPs the methylated form of C/EBPα promotes the development of granulocytes, while the unmethylated form drives monocyte formation, consistent with observations with the Carm1 knockout model (*Greenblatt et al., 2018*). However, further research is needed to determine whether the two forms of C/EBPα exhibit mutually exclusive activities or distinct but overlapping functions.

A role of TF methylation by Carm1 in cell fate directionality has also been described for muscle differentiation of satellite stem cells. Here, asymmetric di-methylation of four arginines in Pax7 results in the recruitment of MLL1/2, activation of Myf5 and subsequent muscle cell specification (*Chang et al., 2018*; *Kawabe et al., 2012*). Another example is the requirement of Carm1 for early embryo development, which has been associated with the methylation of H3R26 and BAF155 and have been proposed as drivers of the process (*Torres-Padilla et al., 2007*; *Panamarova et al., 2016*). However, it seems possible that there are so far unknown TFs targeted by Carm1 that might play a key role.

In summary, our studies about how a mutant of C/EBPα accelerates BMT has provided substantial new insights into the sequential steps occurring during a TF-driven cell fate conversion. Importantly, they suggest that the balance of two forms of a single TF - one post-translationally modified and the other not - can determine the directionality of a binary cell fate decision. This finding challenges the current view that the relative levels of antagonistic lineage-instructive TFs are primarily decisive for cell fate decisions (*Radomska et al., 1998*; *Graf and Enver, 2009*; *Torcal Garcia and Graf, 2021*). Enzyme-mediated TF modifications that accelerate the speed of transdifferentiation, as described here, may apply more broadly to cell fate decisions during development and differentiation. This might apply regardless of whether these decisions occur gradually, such as in hematopoiesis (*Velten et al., 2017*), or abruptly, such as in neuronal differentiation (*Konstantinides et al., 2022*). Further explorations about the relationship between the velocity of a cell fate transition and lineage choice should provide a deeper understanding of the mechanisms that underly developmental disorders, including certain forms of cancer.

## Materials and methods

### Mice

As a source for the primary B cells used in our experiments, we used C57BL/6J mice. The number of female and male mice used was balanced. Mice were housed in standard cages under 12 hr light-dark cycles and fed ad libitum with a standard chow diet. All experiments were approved by the Ethics Committee of the Barcelona Biomedical Research Park (PRBB) and performed according to Spanish and European legislation.

### Primary cells, cell lines, and cell cultures

Primary B cells were isolated from the bone marrow with a monoclonal antibody to Cd19 (BD Biosciences, Cat#553784) using MACS sorting technology (Miltenyi Biotech) as previously described (*Di Stefano et al., 2016*). They were cultured on gelatinized plates containing S17 feeder cells in RPMI culture medium (Gibco, Cat#12633012) containing 20%-FBS (Gibco, Cat#10270-106), 100 U/mL penicillin- 100 ng/mL streptomycin (Gibco, Cat#15140122), 2 mM L-glutamine (Gibco, Cat#25030081) and 0.1 mM 2-mercaptoethanol (Invitrogen, Cat#31350010) (further addressed as mouse B cell medium), which was further supplemented with 10 ng/mL of IL-7 (Peprotech, Cat#217-17). B cells expanded for 3 days correspond to pre-B cells. 293T human embryonic kidney cells, NIH3T3 fibroblasts (ATCC #173), RAW 2647 macrophages (ATCC #TIB-71), and mouse embryo fibroblasts were cultured in 10% FBS (Gibco, Cat#10270-106) DMEM (Gibco, Cat#12491015) medium. The final culture medium also contained 100 U/mL penicillin and 100 ng/mL streptomycin (Gibco, Cat#15140122), 2 mM L-glutamine (Gibco, Cat#25030081), and 0.1 mM 2-mercaptoethanol (Invitrogen, Cat#31350010) (further addressed as DMEM complete medium). The human BLaER1 B cell line was validated to have originated from the RCH-ACV cell line by STR (DSMZ) and contains the t(1,19) translocation based on HiC, as expected. The line and its derivatives were grown in RPMI culture medium (Gibco, Cat#22400089) containing 20%-FBS (10270-106, Gibco) (further addressed as human B cell medium). All cell lines are mycoplasma free based on periodic broad-spectrum mycoplasma testing.

### Induction of mouse BMT

Induction of transdifferentiation of primary B cells (also referred to as B cells) into macrophages was performed as previously described (*Xie et al., 2004*). Briefly, isolated B cells were infected with C/EBPα-ER-hCD4 retrovirus, plated at 500 cells/cm$^2$ in gelatinized plates (12 wells) onto mitomycin-C (Sigma, Cat#M0503)-treated MEFs (10 µg/mL mitomycin-C for 3 hr to inactivate MEFs). Cells were transdifferentiated in mouse B cell medium, supplemented with 10 ng/mL each of IL-7 (Peprotech,

Cat#217-17), IL-3 (Peprotech, Cat#213-13), FLT-3 (Peprotech, Cat#250-31), mCSF-1 (Peprotech, Cat#315-03B), mSCF (Peprotech, Cat#250-03). One hundred nM 17b-estradiol (E2) (Merck Millipore, Cat#3301) was used to shuttle C/EBPαER into the nucleus. B cell culture medium without IL-7 was renewed every 2 days.

## Induction of mouse fibroblast to macrophage transdifferentiation

Fibroblast transdifferentiation into macrophage experiments were performed as previously described (*Feng et al., 2008*). Briefly, NIH 3T3 fibroblasts were infected with C/EBPα-ER-IRES-hCD4 wild-type or mutant retrovirus, hCD4 positive cells sorted and a cell line for each construct. Cells were plated at 200,000 cells/mL in gelatinized six-well plates and infected with PU.1ΔPEST-IRES-GFP retrovirus. After 24 hr cells were re-plated at 30,000 cells/mL in gelatinized 24-well plates in DMEM complete medium supplemented with IL-3 (Peprotech, Cat#213-13) mCSF-1 (Peprotech, Cat#315-03B) and 100 nM E2 to induce C/EBPα, followed by FACS analysis for Mac-1 expression.

## Induction of human BMT

C/EBPα-induced transdifferentiation of BLaER and RCH-ACV cell lines was performed as previously described (*Rapino et al., 2013*). Briefly, RCH-ACV cells grown in human B cell medium were infected with C/EBPα-ER-IRES-GFP retroviruses (wild-type or mutant) and single GFP-positive cells sorted to generate BLaER2 and BLaER2-A lines. These lines were then infected with rtTA-puromycin retroviruses and selected with 1 µg/mL of puromycin for 1 week. Resistant cells were further infected with pHAGE-TetO-Carm1-IRES-dTomato lentiviruses. Cells were induced with 2 µg/mL of Dox (Sigma, Cat#D9891), tomato-positive cells sorted, and single-cell-derived lines RRC3 and RAC1 established. For the induction of transdifferentiation cells grown in human B cell medium supplemented with 10 ng/mL each of IL-3 (Peprotech, Cat#200-03), CSF-1 (Peprotech, Cat#315-03B) were treated with 2.5 µM 4-OHT (Sigma, Cat#H7904).

## Hematopoietic colony forming assay

Bone marrow-derived GMPs from C57BL/6J mice were isolated by FACS, cultured in Methocult GF M3434 (03434, Stem Cell Technologies) for 14 days and scored microscopically.

## Cell transfection

293T cells were transfected by using either calcium phosphate (see below) or polyethylenimine according to the manufacturer's protocol (Polysciences, Cat#24765-2).

## Lentivirus production and infection

Lentiviruses were produced after transfecting 293T cells with 20 µg of lentiviral vector plus 6 µg of pCMV-VSV-G, 15 µg of pCMVDR-8.91, using calcium phosphate. Briefly, calcium phosphate-DNA precipitates were prepared by pooling the three plasmids in a 2.5 M $CaCl_2$ aqueous solution. While vortexing, one volume of HBS 2× (HEPES-buffered saline solution pH = 7.05, 280 mM NaCl, 0.05 M HEPES, and 1.5 mM $Na_2HPO_4$) was added dropwise to an equal volume of the calcium phosphate-DNA solution. The mixture was incubated for 15 min at room temperature and added dropwise to 293T cells grown in DMEM complete medium onto gelatin-coated 100 mm dishes. After 8 hr of incubation at 37°C, the transfection medium was replaced with fresh medium and the supernatant collected after 24 hr. The medium was replaced again, and a second round of supernatant was collected after another 24 hr and mixed with the previous batch. The combined supernatants were centrifuged for 5 min at 300 relative centrifugal force (rcf) and filtered through 0.45 µm strainers to remove cell debris. Lentiviral particles were then concentrated by centrifugation for 2 hr at 20,000 relative centrifugal force (rcf) (Beckman Coulter, Optima L-100K) in round bottom polypropylene tubes (Beckman Coulter, Cat#326823). After discarding the supernatants, the lentiviral pellets obtained from one 150 mm dish were thoroughly re-suspended in 80 µL of PBS. $10^6$ fresh cells were then collected in 900 µL of the respective culture medium and 10 µL of lentiviral suspension were added. Subsequently, the virus-cell mixture was centrifuged at 1000 rcf for 2 hr at 32°C (Beckman Coulter, Allegra X-30R). Infected cells were then cultured as described above and subsequently FACS-sorted for the establishment of clonal cell lines.

## Retrovirus production and infection

For the production of retroviruses used to infect mouse cells (*Bussmann et al., 2009*) we transfected platinum E cells (Cell Biolabs, Cat#RV-101) with the respective constructs. Platinum A cells (Cell Biolabs, Cat#RV-102) were transfected for use in the infection of human cells as previously described (*Di Stefano et al., 2014*).

## Carm1 inhibition experiments

To inhibit Carm1 enzymatic activity we used TP064 (Bio-Techne RD Systems, Bristol, UK) as previously described (*Nakayama et al., 2018*). For experiments with B cells, these were pre-incubated with 5 µM of TP046 24 hr prior to induction with E2, and treatment with the inhibitor continued during the time of induction. For the colony forming assay with GMPs, 1–10 µM of TP064 was added to the medium at the time of plating.

## Cell purification

Mouse bone marrow cell extraction was performed as previously described (*Di Stefano et al., 2014*). Briefly, femurs and tibias of C57BL/6J mice were extracted and crushed on a mortar in PBS supplemented with 4%FBS and 2 mM EDTA and filtered through 0.45 µm strainers (Merck Millipore, Cat#SLHV033RB). For B cells, bone marrow-derived cells were incubated with sequentially 0.1 µg per 1 million cells of both Fc block and CD19-biotin antibody for 10 and 20 min, respectively, followed by 10 µL of magnetic streptavidin microbeads (Miltenyi, Cat#130-048-101) for an additional 20 min. CD19+ cells were sorted using LS columns (Miltenyi, Cat#130-042-401). For BMT CD19+ B cells were infected with C/EBPα-ER-IRES-hCD4 (wild-type and mutants) and cultured over MEF feeder cells for 4 days. Cultured B cells were incubated sequentially with 0.1 µg per 1 million cells of both Fc block and hCD4-biotin antibody for 10 and 20 min, respectively, followed by 10 µL of magnetic streptavidin microbeads (Miltenyi, Cat#130-048-101) for an additional 20 min. hCD4+ cells were enriched with LS columns (Miltenyi, Cat#130-042-401).

For GMPs, bone marrow cells were lineage-depleted using a Lineage Cell Depletion Kit (Miltenyi, Cat#130-090-858). Lineage negative cells were then incubated with CD34-APC, cKit-APC-Cy7, Sca1-PE-Cy7, and CD16/32-FITC for 1.5 hr. Sca1- cKit+ CD34+ CD16/32+ cells (GMPs) were sorted using either FACS Aria or Influx cell sorters. For granulocytes and monocytes, bone marrow-derived cells were incubated sequentially with 0.1 µg per 1 million cells of both Fc block and Mac1-biotin antibody for 10 and 20 min, respectively, followed by 10 µL of magnetic streptavidin beads (Miltenyi, Cat#130-048-101) for an additional 20 min. Mac1+ cells were sorted using LS columns (Miltenyi, Cat#130-042-401) and incubated with Mac1-PE and Ly6g-APC for 20 min. Mac1+ Ly6g- (monocytes) and Mac1+ Ly6g+ cells (granulocytes) were sorted using either FACS Aria or Influx cell sorters.

For 3T3 NIH fibroblasts cells infected with C/EBPα-ER-IRES-hCD4 (WT and R35A) were incubated with 0.1 µg per 1 million cells of both Fc block and hCD4-biotin antibody (BD Pharmingen, Cat#555347) for 10 and 20 min, respectively, followed by 10 µL of magnetic streptavidin beads (Miltenyi, Cat#130-048-101) for an additional 20 min. hCD4+ cells were purified using LS columns (Miltenyi, Cat#130-042-401).

For B lymphoblastic leukemia cells (RCH-ACV) cells stably infected with C/EBPα-ERT2-IRES-GFP, rtTA-Puro and TetO-Carm1-IRES-TdTomato were induced with 1 µg/mL of Dox (Sigma-Aldrich, Cat#D9891). GFP+ and TdTomato+ cells were single cell-sorted using either FACS Aria or Influx cell sorters.

In co-cultures between B cells and feeder cells, non-adherent cells were collected, joined with trypsinized adherent cells, and centrifuged at 300 rcf for 5 min. Cells were re-suspended in 100 µL PBS containing 1 µg/mL of mouse Fc block for 10 min. Conjugated primary antibodies were added to the blocking solution and cells were further incubated at 4°C in the dark for 20 min. Cells were washed with additional 1 mL of PBS and centrifuged at 300 rcf for 5 min. The supernatant was discarded and cells were re-suspended in 500 µL of PBS containing 5 µg/mL of DAPI. Samples were processed in a FACS analyzer (LSR II, BD; Fortessa, BD) with DiVa software and data analyzed using FlowJo software. Antibodies used for cell sorting and flow cytometry are listed in *Supplementary file 3*.

## Phagocytosis assay

After BMT, cells were removed from feeder cells through differential adherence to tissue culture dishes for 40 min. Around 200,000 of the resulting B cells (or induced macrophages) were plated in each well of a 24-well plate containing 0.01% poly-L-lysine-treated coverslips (Corning, Cat#354085) in 10% FBS-DMEM supplemented with IL-3 (Peprotech, Cat#213-13), mCSF-1 (Peprotech, Cat#315-03B) and cultured at 37°C overnight in the presence of 1:1000 diluted blue fluorescent carboxylated microspheres (Fluoresbrite, Cat#17458-10). Cells were centrifuged at 1000 rcf for 5 min to improve attachment to the coverslips. The supernatant was removed and the cells were washed once with PBS.

For fixation, 4% PFA was added to the wells for 20 min, cells were washed twice with PBS and cell membranes permeabilized with 0.1% Triton X-100 PBS (0.1% PBST) for 15 min at room temperature. Cells were blocked using 0.1% PBST with 3% bovine serum albumin (BSA) for 30–45 min. Cells were washed twice in PBS. Actin filaments were subsequently stained with 1:100 diluted red phalloidin (Alexa Fluor 568, Thermo Fisher Scientific, Cat#A12380) while DNA was stained with a 1:500 diluted yellow probe (Quant-iT PicoGreen dsDNA Assay Kit, Thermo Fisher Scientific, Cat#P7589). Cells were incubated with the two dyes in 0.1% PBST containing 1% BSA at room temperature for 1 hr in the dark and washed twice with PBS afterward. Coverslips carrying the attached cells in the well were then recovered with tweezers and mounted upside-down onto a charged glass slide containing a 14 µL drop of mounting medium (7 µL Dako+7 µL 0.1% PBST). Coverslips were sealed with nail polish and imaged in a Leica TCS SPE inverted confocal microscope.

## Proximity labeling of protein interactions by mass spectrometry

C/EBPα- and C/EBPβ-deficient mouse v-abl-transformed B cells were transduced with pRetroX vectors containing Dox-inducible wild-type C/EBPα or R35A C/EBPα constructs C-terminally fused to TurboID or only TurboID ligase as a control (*Branon et al., 2018*). Cells were selected with 1 µg/mL puromycin for 2 weeks in DMEM supplemented with 10% charcoal-treated FBS, 1% penicillin/streptomycin, 1% HEPES, and 50 mM β-mercaptoethanol. One µg/mL Dox was added for 6 hr and proximity labeling performed by addition of 500 µM biotin for 30 min. Cells were then harvested by centrifugation, washed three times in ice-cold PBS, lysed, biotinylated proteins pulled down and subjected to LC-MS/MS as previously described (*Ramberger et al., 2021b*).

## Analysis of mass spectrometry data

Raw data was analyzed using MaxQuant (version 1.6.3.3) against mouse proteome (Uniprot, 2019 version) with isoforms using standard settings. Additional options included fast LFQ, 'match between run', and lysine biotinylation (*Cox et al., 2014*). The data were analyzed using R-statistical software package (version 3.6.1). The protein files from MaxQuant were filtered for contaminants, reverse hits and proteins identified only by site. Intensity values were log2 transformed and filtered to allow only proteins with valid values in all four replicates in at least one of the experimental groups. Remaining missing values were imputed from normal distribution with width of 0.3× standard deviation and downshift of 1.8 from an observed mean. Significant interactors were defined as >2-fold enriched against TurboID and no-Dox controls and to pass the significance cutoff (adj. p-value <0.05) in both t-tests against the two controls. Multiple testing correction was applied using the Benjamini-Hochberg procedure. Differential interactors were scored based on the mean log2FC (WT/R35A)±3 s.d. cutoff. Data from each experimental group (C/EBPα$^{WT}$ and C/EBPα$^{R35A}$ at 6 hpi) were obtained from four replicates.

## Proximity ligation assay

PLA was performed using Duolink Orange Kit (Sigma-Aldrich, Cat#DUO92007). Briefly, after sorting or culturing desired cell populations, 8000–100,000 cells per well were seeded into 24-well plates containing 0.01% poly-L-lysine (Sigma) treated coverslips in appropriate medium, centrifuged at 1000 × *g* for 5 min and fixed with 4% PFA for 15 min. Subsequent steps were performed according to the kit's protocol with antibody concentrations identical to those used for immunofluorescence. Coverslips were mounted using Fluoroshield mounting medium with DAPI (Abcam, Cat#ab104139) and imaged in a Leica TCS SPE confocal microscope.

## Immunostaining by flow cytometry

After antibody staining of cell surface markers, cells were fixed in 4% BSA for 10 min at room temperature in a rotating wheel. Fixation was stopped with two washes in PBS. Cells were permeabilized in

0.1% PBST at room temperature in a rotating wheel for 10 min. Cells were blocked using 0.1% PBST with 3% BSA for 30–45 min. Cells were washed twice in PBS. Cells were incubated with primary antibodies and secondary antibodies diluted at the stated concentrations in 0.1% PBST with 1% BSA for 2 and 1 hr, respectively, with two washes in PBS in between and after. Cells were resuspended in PBS and processed in a FACS analyzer (LSR II, BD; Fortessa, BD) with DiVa software and data analyzed using FlowJo software.

Antibodies used for cell surface immunofluorescence and intracellular staining for flow cytometry are listed in *Supplementary file 1*.

## Protein extraction, immunoprecipitation, and western blotting

Preparation of whole cell lysates and immunoprecipitation of wild-type or mutant C/EBPα proteins were performed as previously described (*Kowenz-Leutz et al., 2010*). Briefly, cells were lysed (20 mM HEPES pH 7.8, 150 mM NaCl, 1 mM EDTA pH 8, 10 mM MgCl$_2$, 0,1% Triton X-100, 10% glycerol, protease inhibitor cocktail [Merck], 1 mM DTT, 1 mM PEFA bloc [Böhringer]). Immunoprecipitation was performed with appropriate antibodies as indicated for 2 hr at 4°C. Immunoprecipitated proteins were collected on Protein A Dynabeads (Invitrogen, Cat#100001D) or Protein-G Dynabeads (Invitrogen, Cat#10004D), separated by SDS-PAGE (Mini PROTEAN TGX, 4–15%, Bio-Rad #5671084). For western blotting, samples were loaded in 10% Mini-PROTEAN TGX gels (Bio-Rad) and resolved by electrophoresis in running buffer (*Supplementary file 2*). Protein samples were transferred to a methanol pre-activated PVDF membrane (Bio-Rad, Cat#1620177) by running them in transfer buffer (TBS) (*Supplementary file 2*) for 1 hr at 300 mA and 4°C. Membranes were rinsed in milliQ water and protein transfer was checked by Ponceau staining (Sigma). Transferred membranes were washed once with TBS and three times with TBS-Tween (TBST) followed by 5% milk in TBST for 45 min. Membranes were then incubated with primary antibodies (*Supplementary file 3*) in 5% milk TBST, rotating overnight at 4°C, then washed three times with TBST followed by incubation with the secondary antibodies conjugated to horseradish peroxidase in 5% milk TBST for 1 hr. After three TBST washes, proteins were detected using enhanced chemiluminescence reagents (Amersham ECL Prime Western Blotting detection) in an Amersham Imager 600 analyzer or visualized by ECL (GE Healthcare, UK). Quantification of band intensity from scanned blots was performed with Fiji software.

## Electrophoretic mobility shift assay

Nuclear extracts were prepared from transfected 293T cells by a mininuclear extract protocol (*Schreiber et al., 1989*). EMSA were performed as previously described (*Kowenz-Leutz et al., 1994*) using double-stranded IRDye Oligonucleotides containing a C/EBP-binding site: IRD800-GACACTGG ATTGCGCAATAGGCTC and IRD800-GAGCCTATTGCGCAATCCAGTGTC (Metabion). Briefly, binding reactions with nuclear extracts (2.5 µg) and double-stranded IRD800 oligos (20 pmol) were incubated for 15 min on ice, orange loading dye (Li-Cor, Cat#P/N 927-10100) was added and protein-DNA complexes were separated on a 5% native polyacrylamide gel in 0.5× TBE at 25 mA at room temperature. EMSA results were visualized and quantified (Odyssey scanner, Licor, channel 800 nm).

## In vitro protein methylation assay

Methylation of peptides (PSL, Heidelberg, Germany, *Supplementary file 4*) was performed using the bioluminescence-based MTase-Glo Assay (Promega, Cat#V7601) according to the manufacturer's protocol. Assay conditions: 200 ng of enzyme was incubated with 5 µM peptide, 10 µM *S*-adenosyl-L-(methyl)-methionine as methyl donor (SAM) and 6× methyltransferse-Glo reagent at 23°C for 60 min. *S*-adenosylhomocysteine generated during the reaction was converted to ADP as a proportional reaction product dependent of substrate methylation by the enzymes. Subsequent incubation with the Methyltransferase-Glo Detection Solution at 23°C for 30 min converts ADP to ATP that is used in a luciferase/luciferin-based reaction and determined as relative light units in a Berthold luminometer (*Hsiao et al., 2016*).

## RNA-sequencing

RNA was extracted with a miRNeasy mini kit (217004, QIAGEN), quantified with a NanoDrop spectrophotometer and its quality examined in a fragment Bioanalyzer (Aligent 2100 Bioanalyzer DNA 7500 assay). CDNA was synthesized with a High-Capacity RNA-to-CDNA kit (4387406, Applied Biosystems).

For RNA-seq, libraries were prepared with a TruSeq Stranded mRNA Library Preparation Kit (Illumina) followed by single-end sequencing (50 bp) on a HiSeq2500 instrument (Illumina), obtaining at least 40 million reads per sample.

Quality control of FASTQ reads was performed using FastQC version v.0.11.3. Reads were mapped aligned to the mm10 genome using STAR version 2.5.0a (*Dobin et al., 2013*). Gene expression was quantified using STAR (--quantMode GeneCounts). Normalized counts and differential gene expression analysis were carried out using DESeq2 version 1.14.1 (*Love et al., 2014*). For each transdifferentiation experiment, timepoint 0 hr was set as a reference point and any gene that exhibited a statistically significant change in expression (log2FC ≥0.5849625 and p-value ≤0.05) at a later timepoint was isolated. For PCA, log2 DESeq2 normalized counts of DEGs averaged across replicates were used. The R prcomp() command with scale = T was used. Pheatmap version 1.0.12 was used to visualize changes in gene expression for all the isolated DEGs with the following clustering options: clustering_distance_rows='correlation', clustering_method='ward.D2', scale='row'.

DEGs were determined for each timepoint as described in the 'Materials and methods'. The union of identified DEGs in the WT and R35A systems per timepoint were used to generate scatterplots depicting the log2FC changes of the aforementioned genes for each transdifferentiation system. A regression line, colored in red, was fit for each scatterplot using the geom_smooth(method = lm) R command. The identity line (y=x line) is depicted in green. The spearman correlation coefficient (cor(method = 'spearman') function in R) and the number of DEGs are also depicted per scatterplot.

## GO analyses

Functional analyses by GO were performed with the R package 'g: profiler2' version 0.2.0 (*Raudvere et al., 2019*). Baloonplots depict all pathways associated with a specific keyword that were found enriched in at least one cluster. Metaplots for each cluster depict the average log2FC values of genes per timepoint and per cluster. Shaded background corresponds to the mean values ± 1.644854 s.d. Gene expression analysis of signature genes was performed using the individual values of the B cell genes *Pax5, Ebf1, Foxo1, Ikzf1, Rag1, Rag2, Bcl11a, Spib, Ikzf3, CD2, CD19, Igll1, Vpreb1, Vpreb2, Vpreb3, Pou2a1, Blk, CD79a, CD79b, Lef1* and the macrophage genes *C1qc, Fcer1g, Sell, Ccr1, Mitf, Tlr2, Csf1r, Trem2, Fam20c, Adam8, Batf2, Fes, Itgam, Ccl3, CD300lf, Tnsf9, Tyrobp, CD14, Ifitm6, Csf3r* , normalized to timepoints o hr for B cell genes and 120 hr for macrophage genes.

## Chromatin accessibility by ATAC-seq

ATAC-seq was performed as published (*Buenrostro et al., 2015*). Briefly, cells were harvested at the mentioned timepoints, feeder-depleted and lysed and 50,000 cells used per condition. Immediately, transposition was performed using Nextera Tn5 Transposase (15027865, Illumina) at 37°C for 30 min. Chromatin was then purified using QIAGEN MinElute PCR Purification Kit (28004, QIAGEN). DNA was then amplified using NEBNext High Fidelity PCR Master Mix (M0541S, New England Biolabs Inc) and barcoded primers. qPCR was performed to determine the optimal number of cycles for each condition to stop amplification prior to saturation. Quality was analyzed by gel electrophoresis and in a fragment Bioanalyzer (Agilent 2100 Bioanalyzer DNA 7500 assay).

Read quality was assessed with FastQC version v.0.11.3. Adaptors were removed using Cutadapt (version 0.4.2_dev) TrimGalore! In paired end mode (--paired –nextera) (*Martin, 2011*). Reads were aligned to the mm10 genome using bowtie2 (v 2.2.4) in paired end mode with standard parameters. Output SAM files were converted to BAM files using samtools (v 0.1.19) (*Li et al., 2009*). BAM files were sorted and indexed using the samtools commands sort and index, respectively. Low-quality reads and reads associated with a not primary or supplementary alignment SAM flag were filtered out using the samtools command 'samtools view -F 2304 -b -q 10'. PCR duplicates were removed with Picard MarkDuplicates (version 2.3.0) with the following options: 'REMOVE_DUPLICATES = true ASSUME_SORTED = true VERBOSITY = WARNING'.

Filtered BAM files were indexed with samtools index and were used as input in the bamCoverage command of deeptools (v3.0.1) (*Ramírez et al., 2016*) in order to generate bigwig files. bamCoverage was used with the options –binSize 1 –normalizeUsing RPGC –effectiveGenomeSize 2150570000 –extendReads –outFileFormat bigwig. In order to call peaks, bam files of each timepoint and experiment were merged using the samtools merge command. Resulting merged bam files were indexed, and peaks were called using MACS2 with the options -f BAMPE –nolambda –nomodel -g mm -q 0.05.

## Determination of differentially accessible ATAC peaks

In order to pinpoint regions of interest, peaks of all timepoints and all experiments were merged using the bedtools suite command bedtools merge. Read counts falling within the merged peak regions were calculated using the Rsubread package and the featurecounts command with the options isPairedEnd = T, strandSpecific = 0, useMetaFeatures = F. For each transdifferentiation experiment, DESeq2 was used in order to compare all timepoints with timepoint 0 hr. Any peak showing a logFC ≥1, an adjusted p-value ≤0.05, and average counts across timepoints ≥5 was termed as a differentially accessible region and was kept for further analyses. The total number of peaks isolated was 91,830. Variance stabilized counts were calculated for the isolated regions using the DESeq2 command varianceStabilizingTransformation and the options 'blind = T'', fitType=''parametric''. Variance stabilized counts were averaged across timepoint replicates by raising them at the power of 2, extracting their average and log2 transforming the resulting mean. PCA was applied to this dataset using the R prcomp command, with 'scale = T'.

To group peaks, PCA was initially applied and PC1 and PC2 values for the 91,830 regions were used in order to arbitrary cluster peaks into three groups depending on the sign of their PC1 and PC2 values. Values for each of the three groups were visualized using the pheatmap package. Visual examination of the three main groups showed different trends: peaks whose accessibility higher at 120 hr (43,429 peaks), lower at 120 hr (36,380 peaks), and higher at 18 hr (12,021 peaks).

## Motif analysis

Peaks from the three different groups were centered and extended 50 bp upstream and downstream. Nucleotide sequences for each centered peak were extracted using bedtools getfasta. Sequences were submitted into MEME-ChIP with the following parameters: -dna -seed 49 -meme-nmotifs 20 -meme-minw 5 -meme-minsites 2 -meme-minw 4 -meme-maxw 12. TOMTOM was run using the output meme.txt file in order to identify matches of known TF motifs to the de novo discovered motifs. For each TOMTOM output a series of additional filtering steps were undertaken:

1. De novo motif sequences need to have ≤ 75% rate for each nucleotide (filtering out repetitive motifs).
2. TOMTOM q-values have to be ≤0.01.
3. The matched TF has to be expressed at least at one timepoint.

## Promoter accessibility analysis

Genomic coordinates of mm10 genes were downloaded from the UCSC table browser (RefSeq genes). A single promoter region was assigned to each gene. The region consisted of 1 kb upstream and downstream of the transcription start site of the largest transcript of each gene. Counts for each timepoint and each transdifferentiation experiment were assigned to each promoter as described above. DESeq2 was used in order to identify differentially accessible promoters as described above with the following differences regarding the cutoffs used: -foldChange ≥ 1.5 and p-value ≤ 0.05. Variance stabilized counts were extracted for each differentially accessible promoter, a mean value per replicate was extracted, and the values were plotted using pheatmap. Promoters were then grouped into eight clusters. Baloonplots depict all pathways associated with a specific keyword that were found enriched in at least one cluster.

For each promoter cluster and each promoter, log2FC changes were extracted by comparing expression levels (DESeq2 normalized counts) of every timepoint with the corresponding timepoint 0 hr of the experiment.

## SMT experiments

Thirty-thousand NIH 3T3 cells inducible for C/EBPα$^{WT}$-Halo or C/EBPα$^{R35A}$-Halo (3T3aER-R and 3T3aER-A, respectively) were seeded in eight-well plates (Lab-Tek 155049), and induced for 6 or 24 hr with 1 μg/mL Dox, with or without prior infection for 24 hr with TetO-FUW-PU.1 lentivirus. Right before imaging, cells were treated with 5 nM of Janelia Fluor 549 (JF549) Halo Tag ligand (a kind gift from Luke Lavis, HHMI) for 15 min. Cells were subsequently washed three times in PBS at 37°C, and GlutaMAX medium (Thermo Fisher, Cat#21063029) was added to each well. All imaging was carried out with a Nanoimager S super resolution microscope (Oxford Nanoimaging Limited) under highly

inclined thin illumination (HILO) conditions (*Tokunaga et al., 2008*). A scientific complementary metal oxide semiconductor camera was used with a 2.3 electrons rms read noise at standard scan, a 100×, 1.49 NA oil immersion objective, and a 561 nm green laser. For imaging experiments, one frame was acquired with a 100 ms exposure time at 10 Hz to measure the intensity of fluorescence of the nuclei, and for SMT experiments, 5000 frames were acquired with an exposure of 10 ms (100 Hz).

### Two-parameter SMT analysis

Quantification and statistical analysis of SMT was performed as previously described (*Lerner et al., 2020*). In brief, TIF stacks SMT movies were analyzed using MATLAB-based SLIMfast script (*Teves et al., 2016*), a modified version of MTT (*Sergé et al., 2008*), with a maximal expected diffusion coefficient (DMax) of 3 $\mu m^2$/s. The SLIMfast output.txt files were reorganized by the homemade csv converter.m MATLAB script (available in *Lerner et al., 2020*) in.csv format for further analysis. The single molecule tracking.csv files (see previous section) were first classified by the homemade SMT_Motion_Classifier.m MATLAB script. Single molecule trajectories (or tracks) with a track duration shorter than five frames were discarded from the analysis. Motion tracks are classified by the script in different groups: tracks with $\alpha \leq 0.7$ were considered as Confined; motion tracks with $0.7 < \alpha < 1$ as Brownian; and motion tracks with $\alpha \geq 1$ as Directed. In addition, the motion tracks showing a behavior similar to a levy-flight (presenting mixed Confined and Directed/Brownian behavior) were detected by the presence of a jump superior to the average jump among the track + a jump threshold of 1.5, and classified as 'Butterfly'. Butterfly motion tracks were segmented into their corresponding Confined and Directed/Brownian sub-trajectories for posterior analysis. As an additional filtering step of confined motions (including confined segments of Butterfly tracks), we defined a jump threshold of 100 nm, to filter out motion tracks with an average frame-to-frame jump size larger than 100 nm. All scripts are publicly available.

### Chromatin immunoprecipitation followed by high-throughput sequencing

To obtain samples for the ChiP-seq experiments biological duplicates of primary B cells expressing either C/EBPα<sup>WT</sup> ER or C/EBPα<sup>R35A</sup> ER were induced for 3 hr with E2 and 100,000 cells per sample processed by ChIPmentation, as described (*Schmidl et al., 2015*) (see also: https://www.nature.com/articles/nmeth.3542). Three µL of anti-C/EBPa antibody (Santa Cruz sc-61 14AA) or anti-PU1 (Santa Cruz sc-352 T21) were used per immunoprecipitation. Tagmentation of immobilized C/EBPa or PU1-enriched chromatin was performed for 5 min at 37 degrees in a 25 µL transposition reaction mix. Libraries were prepared with an Illumina TruSeq ChIP Library Preparation Kit and after quality control on Bioanalyzer sequenced (1 × 50 bp) on an Illumina HiSeq2500 instrument.

ChIP-seq data were processed similarly to ATAC-seq data (refer to 'determination of differentially accessible ATAC peaksanalysis of ATAC-seq data' section) with the following modifications: bamCoverage was used with: `--binSize` 1 `--normalizeUsing` RPKM `--effectiveGenomeSize` 215057000. Peak calling for each experiment was conducted using macs2 callpeak and the options `-f BAMPE -g mm`.

### Identification of differentially occupied peaks

Duplicates from C/EBPα<sup>WT</sup> and C/EBPα<sup>R35A</sup> peaks were merged using bedtools merge. The number of reads per peak were quantified and differentially occupied peaks identified as described in the 'Differentially accessible ATAC-seq peaks' section. Peaks with an average of at least 20 reads, p-adjusted ≤ 0.05 and |logFC| ≥ 1 were classified as differentially occupied peaks. The same process was repeated for PU.1 peaks in the C/EBPα<sup>WT</sup> and C/EBPα<sup>R35A</sup> systems.

### Statistical analyses

Statistical analyses were performed using Prism 9 software. To calculate significance, samples from at least three biologically independent experiments were analyzed. Two biological replicates were used for RNA- and ATAC- sequencing experiments and statistics applied to the expression of a collection of genes. For samples with n≥3, values shown in the figures represent mean ± s.d. Box plots represent median with quartiles and whiskers and individual values are shown. One-way, two-way ANOVA (with the corresponding multiple comparison analyses) and Student's t-tests were applied accordingly.

p-Values appear indicated in each figure only when ≤0.05. In time course experiments, p-values of differences between conditions by two-way ANOVA are shown. In box plots, p-values of each individual timepoint as well as p-values of differences between conditions by two-way ANOVA are shown.

## Newly created materials

All constructs and cell lines listed can be requested from the corresponding authors.

## Acknowledgements

We thank the Graf laboratory members for critical discussions, the Flow Cytometry and Microscopy unit of UPF-CRG for technical assistance, the CRG Genomics core facility for sequencing and Lars Velten for feedback on the manuscript. TG was supported by the Center for Genomic Regulation, Barcelona, the Spanish Ministry of Economy, Industry and Competitiveness, (Plan Estatal PID2019-109354GB-100), AGAUR (SGR 006713) and the 4D-Genome European Research Council Synergy grant. KSZ was supported by the NIH grant R01GM36477. We have used ChatGPT to improve parts of the text.

## Additional information

### Funding

| Funder | Grant reference number | Author |
|---|---|---|
| Agència de Gestió d'Ajuts Universitaris i de Recerca | SGR 006713 | Guillem Torcal Garcia<br>Tian V Tian<br>Luisa De Andres-Aguayo<br>Clara Berenguer<br>Thomas Graf |
| National Institutes of Health | R01GM36477 | Jonathan Lerner<br>Kenneth Zaret |
| Spanish Ministry of Economy, Industry and Competitiveness | PID2019-109354GB-100 | Marcos Plana Carmona |
| European Research Council | Synergy Grant | Guillem Torcal Garcia |

The funders had no role in study design, data collection and interpretation, or the decision to submit the work for publication.

### Author contributions

Guillem Torcal Garcia, Conceptualization, Validation, Investigation, Writing - original draft; Elisabeth Kowenz-Leutz, Tian V Tian, Jonathan Lerner, Investigation, Methodology; Antonis Klonizakis, Data curation, Formal analysis, Investigation; Luisa De Andres-Aguayo, Valeriia Sapozhnikova, Clara Berenguer, Marcos Plana Carmona, Investigation; Maria Vila Casadesus, Data curation, Formal analysis; Romain Bulteau, Mirko Francesconi, Formal analysis; Sandra Peiro, Discussions; Philipp Mertins, Methodology; Kenneth Zaret, Conceptualization; Achim Leutz, Conceptualization, Writing – review and editing; Thomas Graf, Conceptualization, Supervision, Investigation, Writing – review and editing

### Author ORCIDs

Guillem Torcal Garcia ⓘ http://orcid.org/0000-0002-0843-7296
Tian V Tian ⓘ http://orcid.org/0000-0002-9906-0980
Romain Bulteau ⓘ http://orcid.org/0000-0002-9128-8809
Thomas Graf ⓘ http://orcid.org/0000-0003-2774-4117

### Decision letter and Author response

Decision letter https://doi.org/10.7554/eLife.83951.sa1

Author response https://doi.org/10.7554/eLife.83951.sa2

---

## Additional files

### Supplementary files
• Supplementary file 1. Antibodies used for cell surface staining in flow cytometry experiments. The table lists the antibodies used for the experiments and the sources. Related to *Figure 1A, C and D*; *Figure 2A*; *Figure 1—figure supplement 1A* and *Figure 7—figure supplement 2G,H*

• Supplementary file 2. Materials used to prepare buffers for western blot experiments. The table lists the chemical reagents used for western blot experiments. (*Figures 3C; and 7A, B, C*; *Figure 4—figure supplement 1C*; *Figure 7—figure supplement 2A,D,E*).

• Supplementary file 3. Antibodies used for western blot experiments. The table lists the antibodies used for western blot experiments and their sources (*Figures 3C; and 7A, B, C*; *Figure 4—figure supplement 1C*; *Figure 7—figure supplement 2A, D, E*).

• Supplementary file 4. Peptides used for in vitro methylation experiments. The table lists the peptides used for the in vitro Carm1 methylation experiments (*Figure 7—figure supplement 2C*).

• MDAR checklist

### Data availability
The GEO accession number is GSE204746 (https://www.ncbi.nlm.nih.gov/geo/query/acc.cgi?acc=GSE204746).

The following dataset was generated:

| Author(s) | Year | Dataset title | Dataset URL | Database and Identifier |
|---|---|---|---|---|
| Garcia GT, Kowenz-Leutz E, Tian TV, Klonizakis A, Lerner J, Andrés-Aguayo L, Sapozhnikova V, Berenguer C, Plana-Carmona M, Vila-Casadesús M, Bulteau R, Francesconi M, Peiró S, Mertins P, Zaret KS, Leutz A, Graf T | 2023 | Arginine methylation of C/EBPα controls the speed of immune cell transdifferentiation | https://www.ncbi.nlm.nih.gov/geo/query/acc.cgi?acc=GSE204746 | NCBI Gene Expression Omnibus, GSE204746 |

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
