## [Editor Report]

This important manuscript describes how the methylation of a single arginine residue in a transcription factor, C/EBPα, can alter the dynamics of cell fate transition. The study provides one of the most striking examples of transcription factor regulation by methylation and is well-executed, with compelling evidence to support the authors' claims. These findings substantially advance our understanding of a major research question.

---

## [Decision Letter]

**Decision letter after peer review:**

Thank you for submitting your article "Carm1 regulates the speed of C/EBPa-induced transdifferentiation by a cofactor stealing mechanism" for consideration by *eLife*. Your article has been reviewed by 2 peer reviewers, and the evaluation has been overseen by a Reviewing Editor and Edward Morrisey as the Senior Editor. The following individual involved in the review of your submission has agreed to reveal their identity: Joanna Wysocka (Reviewer #2).

Essential revisions:

1) Provide additional quantitative evidence for C/EBPalpha and PU.1 interactions as indicated by the reviewers.

2) Address clarification points raised by each reviewer.

*Reviewer #1 (Recommendations for the authors):*

This paper, from a pioneering group in the fields of molecular hematopoiesis and cell fate reprogramming, strongly establishes the ability of the S35A mutant C/EBPalpha to accelerate BMT to remarkable speed. It also clearly identifies Carm1 as the key methyltransferase responsible for the impact of Arg35 in the function of C/EBPalpha. These are already very interesting and firmly-demonstrated results. The main issue with the paper in its current form is that there is an additional claim about how the S35A mutant is interacting with PU.1 to cause these effects, and that claim is much less clearly supported. Conversion of common target sites from closed to open status, as shown in Figure 4, is not the same as relocating a partner factor to qualitatively different sites across the genome.

The paper could be published with this claim converted to a suggestion, based on the current data, or it could be published in a higher-impact form if additional data could be provided to demonstrate the relocation more directly. The authors would be more expert about the logistics of the experiment, but it seems that a direct ChIP-seq-based comparison should be feasible and powerful for the argument of the paper.

The semiquantitative analysis of S35A binding to PU.1 in vitro suggests a ~2-fold increase in binding, though this would be a stronger interpretation if the experiment were carried out under linear conditions. But if this modest difference is really the basis of the huge increase in reprogramming speed, it suggests that by 1 hr of C/EBPalpha S35A exposure, PU.1 should already have moved away from B cell sites to myeloid sites that it does not occupy in response to WT C/EBPalpha until after more than 6 hr. This would make a very exciting general point about the impact of small affinity differences on global transcription factor redistribution if results like this were found. This kinetic prediction would be interesting to build into the design of the ChIP-seq experiment because then the results could indicate whether PU.1 relocation is fast enough to be a cause of the increased potency of C/EBPalpha S35A mutant effects or just another of its impacts.

*Reviewer #2 (Recommendations for the authors):*

I have several specific suggestions for improvement:

1. In experiments with fibroblast reprogramming, it is important to show that WT and R35A C/EBPα are expressed at equivalent levels in 3T3aER-R and 3T3aER-A cell lines.

2. The description of SMT experiments could be improved. From reading the text, I was confused as to what the authors meant by 'visualizing transcription factor interaction with open and closed chromatin' – how was this defined? Mobility of which protein tracks have been measured in Figure 4D and how was closed and open chromatin defined?

3. Not a big fan of the 'virtual ChIPs' – why not call it for what it is, which is changes in chromatin accessibility at a different subset of regulatory elements, classified based on the C/EBPα and PU.1 binding in B-cells? Figure 4E and F are confusing and imply that C/EBPα and PU.1 binding were actually measured during reprogramming with WT and R35A C/EBPα. In fact, PU.1 ChIP-seq at time points and conditions corresponding to those in Figure 2 would be a valuable addition to the manuscript but are not necessary for publication in this reviewer's opinion.

4. This is a minor point but most of the field refers to 'cofactors' as non-DNA binding coactivators and corepressors (such as Mediator, p300/CBP, etc) recruited to specific genomic sites by TFs. Here, authors refer to PU.1 (a sequence-dependent TF that partners with C/EBPα) as a cofactor; this is not incorrect, but the authors may consider rephrasing, given a different common understanding of the 'cofactor' concept.

---

## [Author Response]

Essential revisions:1) Provide additional quantitative evidence for C/EBPalpha and PU.1 interactions as indicated by the reviewers.2) Address clarification points raised by each reviewer.Reviewer #1 (Recommendations for the authors):This paper, from a pioneering group in the fields of molecular hematopoiesis and cell fate reprogramming, strongly establishes the ability of the S35A mutant C/EBPalpha to accelerate BMT to remarkable speed. It also clearly identifies Carm1 as the key methyltransferase responsible for the impact of Arg35 in the function of C/EBPalpha. These are already very interesting and firmly-demonstrated results. The main issue with the paper in its current form is that there is an additional claim about how the S35A mutant is interacting with PU.1 to cause these effects, and that claim is much less clearly supported. Conversion of common target sites from closed to open status, as shown in Figure 4, is not the same as relocating a partner factor to qualitatively different sites across the genome.The paper could be published with this claim converted to a suggestion, based on the current data, or it could be published in a higher-impact form if additional data could be provided to demonstrate the relocation more directly. The authors would be more expert about the logistics of the experiment, but it seems that a direct ChIP-seq-based comparison should be feasible and powerful for the argument of the paper.

We have now included PU.1 and C/EBPa ChIP-seq experiments, using C/EBPaWT and C/EBPaR35A- induced cells, replacing the virtual ChIP-seq experiments. Integrating the data obtained with our dynamic ATACseq data, the new findings largely support the previously proposed PU.1 redistribution (‘theft’) model. To make the data easier to understand, we now first show the PU.1 and C/EBPa binding to distinct B cell- and macrophage- restricted GREs contained in a single genomic fragment (new Figure 5). The findings nicely visualize how PU.1 becomes redistributed from B-GREs to M-GREs, in a C/EBPa mutant-accelerated manner. We were also happy to see that a genome-wide analysis of the data again shows the accelerated redistribution of PU.1 by C/EBPaR35A (new Figure 6). Finally, the comparison of the ChIP-seq and ATAC-seq data also added more mechanistic detail, such as by revealing that chromatin remodeling of lineage restricted GREs can be uncoupled from the regulation of associated genes.

The semiquantitative analysis of S35A binding to PU.1 in vitro suggests a ~2-fold increase in binding, though this would be a stronger interpretation if the experiment were carried out under linear conditions. But if this modest difference is really the basis of the huge increase in reprogramming speed, it suggests that by 1 hr of C/EBPalpha S35A exposure, PU.1 should already have moved away from B cell sites to myeloid sites that it does not occupy in response to WT C/EBPalpha until after more than 6 hr. This would make a very exciting general point about the impact of small affinity differences on global transcription factor redistribution if results like this were found. This kinetic prediction would be interesting to build into the design of the ChIP-seq experiment because then the results could indicate whether PU.1 relocation is fast enough to be a cause of the increased potency of C/EBPalpha S35A mutant effects or just another of its impacts.

To address the question about the differences in affinities between wild and mutant C/EBPa on the one hand and with PU.1 on the other we have now included a mass spec analysis (by BioID). It nicely reveals PU.1 as the most significantly enriched interactor with C/EBPaR35A, showing a 3.6-fold increase in biotin labeling compared to the wild type protein (Figure 3B). This suggests that the interaction affinity between C/EBPa and its partner is rate limiting for the velocity of the transdifferentiation process, an idea that we have now incorporated into the revised manuscript.

Reviewer #2 (Recommendations for the authors):I have several specific suggestions for improvement:1. In experiments with fibroblast reprogramming, it is important to show that WT and R35A C/EBPα are expressed at equivalent levels in 3T3aER-R and 3T3aER-A cell lines.

We have now performed a cycloheximide treatment experiment with 3T3 cells stably expressing inducible forms of the two proteins (3T3aER-R and 3T3aER-A). The data in Figure S4C show that C/EBPaR35A exhibits a similar stability than WT protein and is expressed at 20-30% lower levels under steady-state conditions, agreeing with the observed slightly lower overall biotin labeling of mutant proteins compared to wild type in the BioID assay (Figure 3B). These findings are also in line with the comparison of the two proteins in Western blots of transfected 293T cells and infected B cells, where the two proteins show similar expression levels (Figure 7C and D). Therefore, the finding that expression of C/EBPaR35A is similar or slightly lower than that of the wild type argues against the possibility that an elevated expression level of the mutant could explain the effects observed.

2. The description of SMT experiments could be improved. From reading the text, I was confused as to what the authors meant by 'visualizing transcription factor interaction with open and closed chromatin' – how was this defined? Mobility of which protein tracks have been measured in Figure 4D and how was closed and open chromatin defined?

We have now carefully re-written this section to better explain the experiment and also included a diagram outlining the experiment (Figure 4B).

3. Not a big fan of the 'virtual ChIPs' – why not call it for what it is, which is changes in chromatin accessibility at a different subset of regulatory elements, classified based on the C/EBPα and PU.1 binding in B-cells? Figure 4E and F are confusing and imply that C/EBPα and PU.1 binding were actually measured during reprogramming with WT and R35A C/EBPα. In fact, PU.1 ChIP-seq at time points and conditions corresponding to those in Figure 2 would be a valuable addition to the manuscript but are not necessary for publication in this reviewer's opinion.

We have now included PU.1 and C/EBPa ChIP-seq experiments, using C/EBPaWT and C/EBPaR35A- induced cells, replacing the virtual ChIP-seq experiments. Integrating the data obtained with our dynamic ATACseq data, the new findings largely support the previously proposed PU.1 redistribution (‘theft’) model. To make the data easier to understand, we now first show the PU.1 and C/EBPa binding to distinct B cell- and macrophage- restricted GREs contained in a single genomic fragment (new Figure 5). The findings nicely visualize how PU.1 becomes redistributed from B-GREs to M-GREs, in a C/EBPa mutant-accelerated manner. We were also happy to see that a genome-wide analysis of the data again shows the accelerated redistribution of PU.1 by C/EBPaR35A (new Figure 6). Finally, the comparison of the ChIP-seq and ATAC-seq data also added more mechanistic detail, such as by revealing that chromatin remodeling of lineage restricted GREs can be uncoupled from the regulation of associated genes.

4. This is a minor point but most of the field refers to 'cofactors' as non-DNA binding coactivators and corepressors (such as Mediator, p300/CBP, etc) recruited to specific genomic sites by TFs. Here, authors refer to PU.1 (a sequence-dependent TF that partners with C/EBPα) as a cofactor; this is not incorrect, but the authors may consider rephrasing, given a different common understanding of the 'cofactor' concept.

We have now replaced the designation of PU.1 as a ‘co-factor’ with ‘TF partner’ of C/EBPa.

Finally, although not requested by the reviewer, we have now addressed the possibility that the effect of the alanine replacement of R35 is mostly due to a change from a charged to a non-charged hydrophobic residue. This is not the case, as a replacement of arginine 35 by the charged amino acid lysine still leads to an accelerated BMT induction (Figure S7).